# Design and implementation of aerobic and ambient CO₂-reduction as an entry-point for enhanced carbon fixation

Ari Satanowski [1,2,6] ✉, Daniel G. Marchal [1,6], Alain Perret[3], Jean-Louis Petit[3], Madeleine Bouzon [3], Volker Döring[3], Ivan Dubois[3], Hai He [1], Edward N. Smith [4], Virginie Pellouin[3], Henrik M. Petri [1], Vittorio Rainaldi [2], Maren Nattermann[1], Simon Burgener [1], Nicole Paczia [1], Jan Zarzycki [1], Matthias Heinemann [4], Arren Bar-Even [2,7] & Tobias J. Erb [1,5] ✉

The direct reduction of CO₂ into one-carbon molecules is key to highly efficient biological CO₂-fixation. However, this strategy is currently restricted to anaerobic organisms and low redox potentials. In this study, we introduce the CORE cycle, a synthetic metabolic pathway that converts CO₂ to formate at aerobic conditions and ambient CO₂ levels, using only NADPH as a reductant. Combining theoretical pathway design and analysis, enzyme bioprospecting and high-throughput screening, modular assembly and adaptive laboratory evolution, we realize the CORE cycle in vivo and demonstrate that the cycle supports growth of *E. coli* by supplementing C1-metabolism and serine biosynthesis from CO₂. We further analyze the theoretical potential of the CORE cycle as a new entry-point for carbon in photorespiration and autotrophy. Overall, our work expands the solution space for biological carbon reduction, offering a promising approach to enhance CO₂ fixation processes such as photosynthesis, and opening avenues for synthetic autotrophy.

The biological conversion of inorganic carbon into biomass is at the core of the global carbon cycle, and key to establish a sustainable (bio) economy, in which the greenhouse gas carbon dioxide (CO₂) serves as the main carbon source. In chemical terms, formation of biomass from CO₂ requires the reduction and condensation of the one-carbon (C1) molecule CO₂ into multi-carbon compounds. This can in principle happen via two different routes: Most CO₂-fixing organisms use "carbon-fixation" routes, in which CO₂ is first incorporated into an acceptor molecule, subsequently reduced and directly converted into a multi-carbon compound through the same pathway[1]. Alternatively, CO₂ can be assimilated through a "reduction-first" strategy[2], in which CO₂ is initially converted into a reduced C1-compound (e.g. formate or

carbon monoxide), which is condensed into multi-carbon compounds in a subsequent phase through downstream reactions.

The reduction-first strategy offers several advantages in terms of energetic efficiency and rate, which makes it highly attractive from a biotechnological point of view[2]. Two of the eight naturally existing CO₂-assimilation pathways follow this strategy, namely the reductive acetyl-CoA pathway (also known as the Wood-Ljungdahl pathway) and the reductive glycine pathway[1,3–6]. However, both routes were exclusively found in organisms that grow anaerobically, in the presence of low-redox potential electron donors (e.g., molecular hydrogen (H₂)), and preferably at high CO₂ partial pressures[5,7]. While microbial strains using these strategies have been developed for industry and can be

[1]Max Planck Institute for Terrestrial Microbiology, Karl-von-Frisch-Str. 10, Marburg, Germany. [2]Max Planck Institute of Molecular Plant Physiology, Am Mühlenberg 1, Potsdam, Germany. [3]Génomique Métabolique, Genoscope, Institut François Jacob, CEA, CNRS, Univ Evry, Université Paris-Saclay, Evry-Courcouronnes, France. [4]Molecular Systems Biology, Groningen Biomolecular Sciences and Biotechnology Institute, University of Groningen, Nijenborgh 7, Groningen, Netherlands. [5]Center for Synthetic Microbiology (SYNMIKRO), Karl-von-Frisch-Straße 14, Marburg, Germany. [6]These authors contributed equally: Ari Satanowski, Daniel G. Marchal. [7]Deceased: Arren Bar-Even. ✉e-mail: ari.satanowski@gmail.com; toerb@mpi-marburg.mpg.de

grown under these special conditions[8], many bioproduction hosts, and especially photosynthetic organisms including crops, are restricted to aerobic environments and atmospheric $CO_2$ levels. This has raised the question of whether reduction-first routes could, in principle, also be realized under aerobic conditions.

A critical challenge to establish an oxygen-tolerant reduction-first $CO_2$-fixation is the initial step, namely reduction of $CO_2$ to formate. This reaction requires a redox potential of roughly −460 mV at ambient $CO_2$-concentrations (400 ppm $CO_2$ and physiological pH of 7 (eQuilibrator[9]; $E_0' = -432$ mV with 1 bar $CO_2$[10]). Aerobic organisms that do not possess low-redox potential electron donors have to rely on the NAD(P)H/NAD(P)$^+$ pair (−320 mV[10]). This constrains the direct reduction of $CO_2$, with an estimated Gibbs energy change ($\Delta G$) of +25 kJ/mol (400 ppm $CO_2$, pH 7 and 1 mM for all other reactants[9]), which is almost impossible to achieve with physiological NADPH/NADP$^+$ ratios[11–18] and additionally comes at very high enzyme cost to reach sufficient flux[18–20]. This general limitation is supported by a recent study, in which aerobic $CO_2$-reduction via an NADH-dependent formate dehydrogenase was successfully implemented in vivo, but only allowed for slow growth at strongly elevated $CO_2$ concentrations >10% (i.e., 0.1 bar, 250× atmospheric level) and required an evolved mutation in the respiratory chain, likely increasing the cellular NADH/NAD$^+$ ratio[21].

In this work, we develop a metabolic architecture that is able to convert $CO_2$ into formate at fully aerobic conditions, and ambient $CO_2$-concentrations. To overcome the thermodynamic and kinetic bottlenecks associated with $CO_2$-reduction, we design several pathways ("mini-cycles") that are energized by ATP-hydrolysis (Fig. 1A, B) and split up the task of $CO_2$-reduction into multiple enzymatic steps with favorable thermodynamic driving forces. We use high-throughput in vitro bioprospecting to identify suitable enzyme candidates for our designs and realize one of these cycles, the CORE cycle, which we successfully assemble in a modular fashion in *Escherichia coli*. Using adaptive laboratory evolution (ALE) allows us to achieve a fully functional CORE cycle that supports growth of *E. coli* by supplying the organism's complete C1-metabolism and serine biosynthesis from $CO_2$. The CORE cycle provides a flexible design that holds the potential to improve the efficiency of natural $CO_2$-fixation pathways, such as photosynthesis (e.g., by augmenting photorespiration[22], Fig. 1D), and—at the same time—also opens the door for establishing synthetic reduction-first autotrophy in the future (Fig. 1C).

## Results
### Design and evaluation of $O_2$-tolerant, NAD(P)H-dependent $CO_2$-reduction pathways
In a theoretical effort, we first designed ten different hypothetical pathways for the ATP-driven, NAD(P)H-dependent conversion of $CO_2$ to formate (Supplementary Figs. 1–10, Supplementary Table 1). These pathways include at least four reactions (Fig. 1B): (i) carboxylation (incorporation of $CO_2$ or bicarbonate), (ii) ATP-hydrolysis, (iii) reduction, and (iv) release of formate. All pathways follow this logic but differ in the number and order of the reactions, the amount of ATP hydrolyzed, the reductant (NADH or NADPH), and the "backbone molecule" (i.e. the acceptor substrate) for the initial $CO_2$ incorporation (Supplementary Table 1).

We evaluated our ten pathway designs based on multiple criteria (Supplementary Table 1). Specifically, we compared their (i) resource requirements (ATP and reducing equivalents), (ii) pathway length (i.e. number of enzymes), (iii) need for complex enzyme co-factors (e.g. adenosylcobalamin (coenzyme B12)) potentially limiting the implementation in organisms such as *E. coli* or plants[23,24], and (iv) number of new-to-nature enzyme reactions[25–27] that would need to be established to realize the cycles. This analysis identified four promising designs that we considered further for implementation. These pathways all required only one new-to-nature reaction as a key step, after which we initially named them: the aspartate-formate-lyase cycle, the reductive

formamide pathway, the oxaloacetate hydrolase cycle, and the β-keto-acid cleavage cycle (detailed in Supplementary Note 1).

### In vitro prototyping of key enzyme activities identifies the CORE cycle
We next focused on establishing the missing new-to-nature reaction for each of the four different cycles. To that end, we performed a literature search to identify and test different enzyme candidates that would be able to catalyze the required novel reaction. While we failed to establish activity for three of the reactions in vitro (aspartate formate lyase, carbamoyl-phosphate reductase, oxaloacetate hydrolase), we could successfully demonstrate the β-keto acid cleavage reaction (see below). Thus, we sought to realize the full β-keto acid cleavage cycle, which we will refer to as the CORE cycle ($CO_2$-reduction cycle; Fig. 1E) in the following.

The CORE cycle achieves a net conversion of $CO_2$ (bicarbonate) to formate at the expense of 1 ATP and reducing power in the form of 1 NADPH. The pathway can also be envisioned in two other "design variants" which differ in their ATP-cost to activate acetoacetate to acetoacetyl-CoA (0–2 ATP, reaction #4 in Fig. 1E; Supplementary Table 1, Supplementary Fig. 1). The CORE cycle first incorporates bicarbonate via carboxylation of acetyl-CoA into malonyl-CoA, followed by reduction of malonyl-CoA to malonate semialdehyde (MSA) through malonyl-CoA reductase (MCR). Next, a β-keto acid cleavage enzyme (BKACE)[28,29] converts the non-native substrate MSA with acetyl-CoA into acetoacetate and formyl-CoA (Fig. 1F). Conversion of formyl-CoA to the final product formate can occur either via hydrolysis (by thioesterases or non-enzymatically[30–32]), or in a more energy-conserving manner by transferring the CoA-moiety from formyl-CoA onto acetoacetate via a CoA-transferase. To close the cycle, acetoacetyl-CoA is cleaved by a thiolase, thus regenerating two molecules of acetyl-CoA, one of which enters another round of the cycle, while the other is required for the BKACE reaction.

### Identification and in-depth characterization of enzyme candidates for the CORE cycle
All reactions of the CORE cycle apart from the proposed BKACE reaction were already known from other natural pathways. To identify suitable enzymes for these reactions, we tested multiple candidates in vitro (Supplementary Table 2). For the carboxylation of acetyl-CoA (reaction #1, Fig. 1E), we identified an engineered propionyl-CoA carboxylase (Pcc_Me_D407I)[33] that is more stable in vitro and less complex compared to many natural acetyl-CoA carboxylases[33]. For the NADPH-dependent reduction of malonyl-CoA to MSA (reaction #2), we considered two options: Archaeal MCRs, and bacterial bi-functional MCRs that catalyze the additional reduction of MSA to 3-hydroxypropionate[34–36]. We tested candidates from both clades (Supplementary Fig. 11A) and decided to use a truncated version of the bi-functional MCR from *Chloroflexus aurantiacus*, that comprises only its C-terminal domain catalyzing the reduction of malonyl-CoA to MSA[37]. For the activation of acetoacetate (reaction #4), we tested four candidates that work either by CoA ligation or CoA transfer. For the β-ketothiolase (reaction #5, acetyl-CoA acetyltransferase), we selected AtoB from *E. coli* (Supplementary Table 2).

To identify candidates that catalyze the new-to-nature BKACE reaction, we leveraged a previously described collection of BKACE proteins[29]. We tested 124 enzymes for the ability to convert MSA and acetyl-CoA into formyl-CoA and acetoacetate (Fig. 1F) and identified more than 20 homologs with the desired activity. We screened these candidates with an endpoint assay (Supplementary Fig. 11B and Supplementary Table 3) and characterized the eight most promising candidates in detail using an assay in which we measured acetyl-CoA consumption, formyl-CoA production and acetoacetate production under substrate-saturated conditions (Fig. 2A, Supplementary Fig. 12).

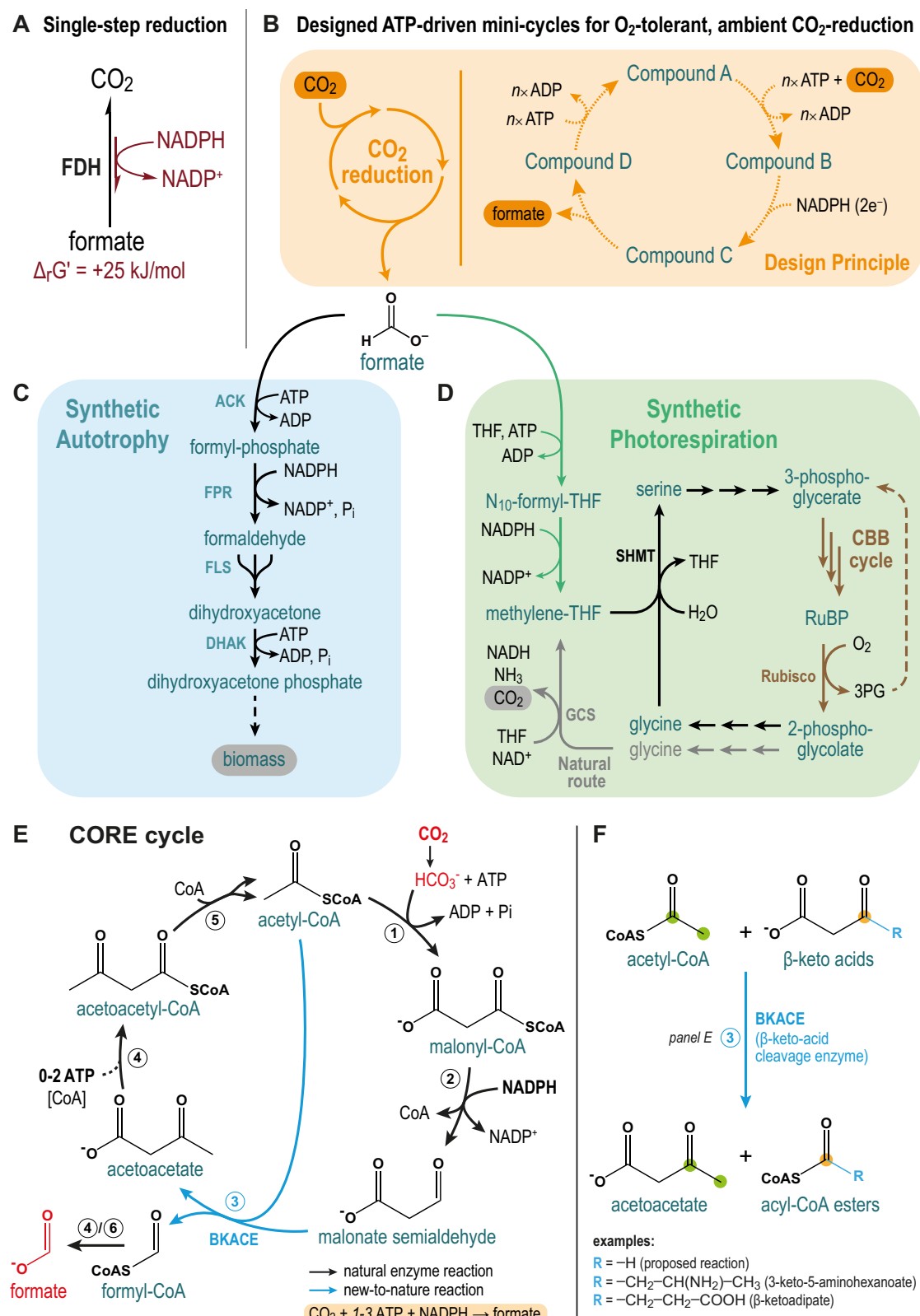

**A  Single-step reduction**

**B  Designed ATP-driven mini-cycles for O₂-tolerant, ambient CO₂-reduction**

**C  Synthetic Autotrophy**

**D  Synthetic Photorespiration**

**E  CORE cycle**

$CO_2 + 1\text{-}3\ ATP + NADPH \rightarrow formate$

**F**

examples:
R = −H (proposed reaction)
R = −CH₂−CH(NH₂)−CH₃ (3-keto-5-aminohexanoate)
R = −CH₂−CH₂−COOH (β-ketoadipate)

In these assays, BKACE15 from *Paracoccus denitrificans* performed best (Supplementary Table 4). The physiological role of this enzyme is unknown. However, the genetic context suggests that BKACE15 is involved in the degradation of aromatic compounds, while experimentally confirmed substrates include β-ketohexanoate, 3,5-dioxohexanoate, 5-hydroxy-β-ketohexanoate and β-ketopentanoate[29]. With acetyl-CoA and MSA, BKACE15 showed a

specific activity of $770 \pm 30$ nmol min⁻¹ mg⁻¹ and strong substrate inhibition above 10 mM MSA. Limiting the analysis to physiologically relevant concentrations in the lower mM range, we determined an apparent $K_m$ value for acetyl-CoA at $0.11 \pm 0.01$ mM, and MSA at $0.83 \pm 0.09$ mM (Figs. 2B, C, Supplementary Table 5). However, we also observed that the enzyme was inhibited by its product acetoacetate with a half-maximal inhibitory concentration ($IC_{50}$) of

**Fig. 1 | ATP-driven CO$_2$-reduction as a novel entry-point for carbon fixation.**
**A** Single-step reduction of CO$_2$ by formate dehydrogenases (FDH) is thermo-dynamically and kinetically constrained under ambient CO$_2$-conditions (see main text). **B** Design principles for the CO$_2$ reduction pathways proposed here. Aerobic conversion of CO$_2$ to formate at ambient CO$_2$ concentrations is pursued by ener-gizing the process with ATP-hydrolysis in short, cyclic pathways. The order and number of the indicated steps varies among the proposed routes, while multiple reactions may also be catalyzed within a single enzyme (e.g. carboxylation and ATP-hydrolysis in the case of biotin-dependent carboxylases). **C** Proposed synthetic autotrophy based on extending the CO$_2$ reduction pathway with a subsequent module to assimilate formate into multi-carbon products, shown for an example route consisting of: ACK (promiscuous) acetate kinase[84], FPR formyl-phosphate reductase[84], FLS formolase[85], DHAK dihydroxyacetone kinase. **D** Proposed syn-thetic photorespiration based on a CO$_2$ reduction pathway, circumventing the inefficient carbon-releasing reactions of natural photorespiration (glycine dec-arboxylation). A more detailed depiction is shown in Supplementary Fig. 26. CBB

cycle Calvin-Benson-Bassham cycle, GCS glycine-cleavage-system, RuBP ribulose bisphosphate, SHMT serine hydroxymethyltransferase, 3PG 3-phosphoglycerate, THF tetrahydrofolate. **E** The CORE cycle incorporates CO$_2$ in the form of bicarbo-nate by acetyl-CoA carboxylation (#1), producing malonyl-CoA that is subsequently reduced (#2) to malonate semialdehyde (MSA). The proposed BKACE reaction (#3) then condenses MSA with acetyl-CoA to produce formyl-CoA and acetoacetate. Formyl-CoA can be used as a CoA donor to activate acetoacetate to acetoacetyl-CoA (#4), releasing formate. Finally, 2 molecules of acetyl-CoA are regenerated from acetoacetyl-CoA by a β-ketothiolase reaction (#5). As an alternative to the CoA-transferase variant, formyl-CoA can be hydrolyzed instead (#6), while acet-oacetate is activated at the expense of additional ATP (via a kinase/ligase/synthe-tase, Supplementary Fig. 1). **F** Generic scheme of the BKACE reaction. The Claisen-like condensation reaction accepts various β-keto acids that react with acetyl-CoA to form an acyl-CoA ester and acetoacetate[28,29,129]. In the reaction proposed here for the CORE cycle, MSA (the smallest β-keto acid) is used to produce formyl-CoA (the smallest CoA-ester).

---

$1.45 \pm 0.16$ mM (Fig. 2D, Supplementary Fig. 13). We noted that the $IC_{50}$ for acetoacetate was in the same range as the $K_m$ value for MSA, thus posing a potential bottleneck, if acetoacetate was not removed quickly downstream of the BKACE reaction.

## Structure-function studies explain the reaction mechanism of BKACE

To understand the catalytic behavior of BKACE15, we obtained struc-tures of the apo-form at 1.45 Å (PDB 8RIO), the enzyme in complex with malonate and free CoA at 1.81 Å (PDB 8RIP), and in complex with acetoacetate and acetyl-CoA at 2.10 Å (PDB 9HNF) (Supplementary Fig. 14). BKACE15 forms a homotetramer where each active site is accommodated in a TIM-barrel fold, and consists of eight β-sheets that are surrounded by eight α-helices (Fig. 2E). The two openings of the TIM barrel are closed off by smaller α-helices. As previously reported for the BKACE from *Candidatus Cloacamonas acidaminovorans*, a close homolog of BKACE15, each active site coordinates a single Zn$^{2+}$ ion[28]. We confirmed a 1:1 zinc binding ratio in BKACE15 using Induc-tively Coupled Plasma Optical Emission Spectroscopy (ICP-OES, Sup-plementary Fig. 15). Each active site also exhibits only one entry channel for both substrates that appears fully occupied upon acetyl-CoA binding (Fig. 2F, G). This suggests a catalytic mechanism, in which MSA must enter the active site first, before acetyl-CoA can bind. After the reaction has taken place, formyl-CoA is released before acet-oacetate can leave the active site. This hypothesis is further corrobo-rated by the observation that acetoacetate is an inhibitor for the reaction with MSA and acetyl-CoA.

Earlier studies had proposed R254 as a catalytic base involved in the reaction (Supplementary Fig. 16)[28,29]. However, our structures with bound CoA-ester prompted us to revisit the nature of the catalytic base and suggest a different mechanism for the reaction with MSA, in which E171 acts as the catalytic base (Fig. 2G, discussed in more detail in Supplementary Note 2, Supplementary Fig. 17). Replacing E171 by leucine abolished BKACE activity, while substitution by a chemically similar but shorter aspartate still showed basal activity (5% of the wild type; Supplementary Table 4), supporting the role of E171 in catalysis.

## Modular implementation of the CORE cycle in vivo: Modules 2 + 4

We next aimed to demonstrate the CORE cycle in vivo. To implement the CORE cycle in *E. coli*, we used a modular strategy, in which parts of the pathway (i.e., "pathway modules") are successively assembled and tested in growth-coupled selection strains[38–46]. For the CORE cycle, we defined four modules (Fig. 3): Module 1 converts one molecule of acetyl-CoA and bicarbonate into MSA. Module 2 consists of the BKACE reaction and produces formyl-CoA and acetoacetate. Module 3 closes the CORE cycle by regenerating acetyl-CoA from acetoacetate. Lastly, module 4 enables growth-coupled selections by converting formate

into the cellular building blocks formyl-tetrahydrofolate (THF), methylene-THF and serine (by condensation of methylene-THF with glycine).

We first constructed an *E. coli* strain that allows for growth-coupled detection of formate (i.e., the CORE cycle product). Guided by previous studies[21,38–40,42–44,47,48], we deleted *serA* and *gcvTHP* (encoding 3-phosphoglycerate dehydrogenase and the glycine cleavage system, respectively), creating a strain that is unable to synthesize serine as well as the intermediates of one-carbon metabolism (methyl-tetra-hydrofolate (THF), methylene-THF and formyl-THF; jointly referred to as C1-THF in the following). In this deletion background, we chromo-somally integrated three genes from *Methylobacterium extorquens* to enable formate-dependent biosynthesis of C1-THF and serine[39,42] (Fig. 4A): *ftfL* (formate-THF ligase), *fchA* (methenyl-THF cyclohy-drolase), and *mtdA* (methylene-THF dehydrogenase). We additionally deleted the *frmRAB* operon (removing formaldehyde oxidation as a potential intracellular source of formate) as well as two endogenous alcohol dehydrogenases that are known to drain the CORE cycle intermediate MSA (Δ*ydfG* Δ*rutE*)[49–52]. The resulting strain is referred to as "C1S-Aux" in the following.

Notably, disruption of serine biosynthesis (Δ*serA*) also abolished canonical glycine biosynthesis in the C1S-Aux strain. Although gly-cine could be generated from threonine cleavage (via *ltaE* or *kbl-tdh*), the corresponding genes are not constitutively expressed[42,53]. Thus, we supplemented small amounts of glycine (2 mM) to minimal media in all experiments. In minimal medium containing a main carbon source (e.g. glycerol), growth of the C1S-Aux strain was strictly dependent on the supplementation of formate (and glycine), allow-ing robust detection of formate concentrations as low as 80 μM (Supplementary Fig. 18). Altogether, these experiments established the formate selection strain and verified CORE cycle module 4 in *E. coli*.

We next tested module 2, which comprised BKACE15 (Fig. 3). To provide MSA as the substrate for the BKACE reaction in vivo, we co-expressed a heterologous β-alanine:pyruvate transaminase (BPT; Uni-Prot Q9I700[54]), which allowed for the intracellular generation of MSA from β-alanine, when this compound was provided in the medium (Fig. 4A). Besides BKACE15, we also tested seven of the well-performing BKACE candidates identified in the in vitro screen, of which only BKACE137 achieved comparable growth rates (Fig. 4B). Growth was strictly β-alanine-dependent and correlated with the supplied β-alanine concentration, suggesting that module 2 of the CORE cycle was active (Fig. 4C). To independently confirm module 2 activity, we used stable isotope tracing with $^{13}C_3$-$^{15}N$-β-alanine as substrate. These experiments showed almost exclusively single-labeled (M + 1) serine and methionine, confirming that the C1-derived atoms of these amino acids were indeed produced through modules 2 + 4 of the CORE cycle (Supplementary Fig. 19).

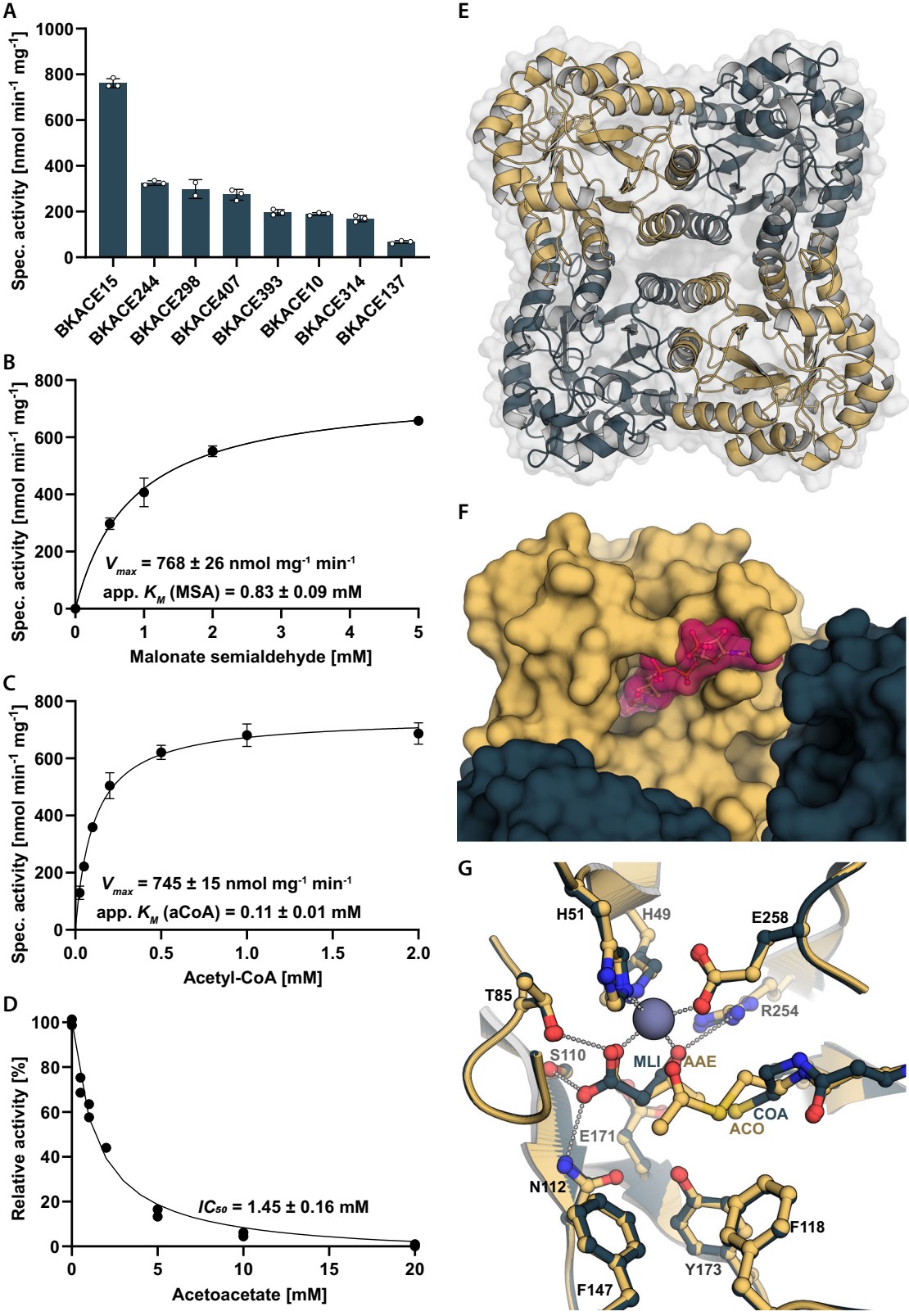

**Modular implementation of the CORE cycle in vivo: Modules 1 + 2 + 4**

Having demonstrated that modules 2 and 4 of the CORE cycle were functional in vivo, we sought to add more pathway modules towards assembling the entire cycle. We focused on reconstituting module 1 next. For the conversion of acetyl-CoA into malonyl-CoA, we relied on endogenous acetyl-CoA carboxylase activity (Fig. 5A), while for the

subsequent reduction of malonyl-CoA to MSA, we expressed the C-terminal domain of the bi-functional MCR from *C. aurantiacus*[37] (see above).

Simultaneous expression of MCR and BKACE15 (from the pCORE plasmid) rescued growth of the CIS-Aux selection strain. Notably, growth was observed on various tested carbon sources including glycerol, D-glucose and L-Lactate (Fig. 5B), indicating that the CORE cycle

**Fig. 2 | Biochemical characterization of BKACE. A** Specific activities of selected BKACE homologs with 4 mM MSA and 1 mM acetyl-CoA. Homologs were chosen from the initial high-throughput screen. Data are presented as individual and mean values ± SD. **B** Michaelis-Menten kinetics for BKACE15 with different malonate semialdehyde concentrations. We excluded concentrations above 5 mM MSA as we observed substrate inhibition with increasing concentrations. Data are presented as mean ± SD. **C** Michaelis-Menten kinetics for BKACE15 with different acetyl-CoA concentrations. Data are presented as mean ± SD. **D** Relative activity of BKACE15 and its inhibition. Measurements were done with 5 mM MSA, 1 mM acetyl-CoA and different concentrations of acetoacetate. Relative activities were calculated by normalizing the data to the activity with 0 mM acetoacetate. Data are presented as mean ± SD. $IC_{50}$: Half maximal inhibitory constant. All measurements were done in duplicates (**D**) or triplicates (**A, B, C**). Quantification of acetoacetate formation (A, B, C) or formyl-CoA formation (**D**) was used to determine enzyme activities. The data were analyzed using nonlinear regression (B, C, D). **E)** Overall structure of BKACE15. The four monomers of the tetrameric complex are indicated by different colors. **F** Active site entry channel of BKACE15 with bound acetyl-CoA (magenta) and acetoacetate (not visible) (PDB 9HNF). The surface representation of acetyl-CoA highlights the complete occupancy of the entry channel. **G** Overlay of BKACE15 active sites of PDB 8RIP (malonate and CoA bound) and PDB 9HNF (acetoacetate and acetyl-CoA bound). Malonate, CoA and the corresponding backbone are shown in dark grey, acetoacetate, acetyl-CoA and the corresponding backbone in yellow, zinc ($Zn^{2+}$) in metal grey, polar interactions in light grey dashes. Substrate labeling is shown in yellow for acetyl-CoA ("ACO") and acetoacetate ("AAE") and in dark grey for CoA ("COA") and malonate ("MLI"). Labeling of foreground residues in black, labeling of background residues in dark grey. Electron density maps of substrates in the active site are shown in Supplementary Fig. 14. Source data are provided as a Source Data file.

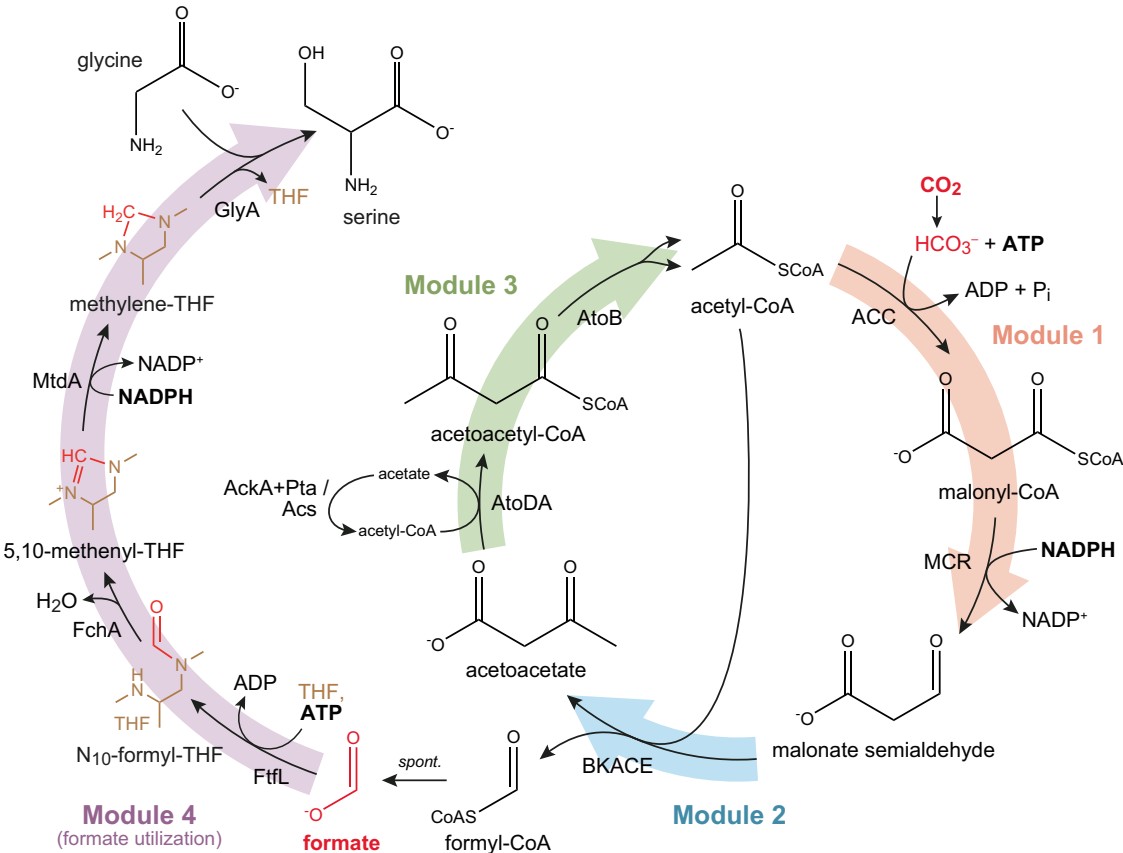

**Fig. 3 | Subdivision of the CORE cycle into pathway modules facilitates stepwise implementation in *E. coli*.** Here, module 1 was established based on endogenous ACC (acetyl-CoA carboxylase) activity and heterologous MCR (malonyl-CoA reductase). Module 2 consists of the BKACE reaction, while implementation of module 3 was pursued based on endogenous activities alone [AtoDA (acetyl-CoA:acetoacetate CoA-transferase); AtoB (thiolase); acetyl-CoA regeneration either via Acs (acetyl-CoA synthetase) or a combination of AckA (acetate kinase) and Pta (phosphotransacetylase)]. The CORE cycle is integrated with subsequent formate assimilation (module 4), which serves as a selectable biosynthesis route to serine and essential one-carbon units of biomass (i.e. formyl-THF, methylene-THF and methyl-THF).

modules 1, 2 and 4 are robust across different steady-state concentrations of acetyl-CoA[17]. Control strains expressing only BKACE15 or MCR (i.e., lacking either module 1 or 2, respectively) failed to grow under the same conditions (Fig. 5B).

Notably, even though *E. coli* is known to grow on acetoacetate utilizing the native *ato* operon[52,55–57] (AtoDA and AtoB comprising the CORE cycle module 3), we could not observe growth of the C1S-Aux + pCORE strain with acetoacetate as the main carbon source. We observed growth only with externally provided formate, which rescues the selection (Fig. 5B). We thus sought to employ adaptive laboratory evolution (ALE) to establish the complete CORE cycle including module 3 in vivo.

### Adaptive laboratory evolution enables operation of the complete CORE cycle

Operating the CORE cycle in vivo requires an intricate balancing of enzyme activities within the cycle and between overlapping pathways. This is particularly apparent for acetyl-CoA, which is a central metabolite in energy conservation (TCA cycle), fatty acid and amino acid metabolism, and also represents a branching point within the CORE

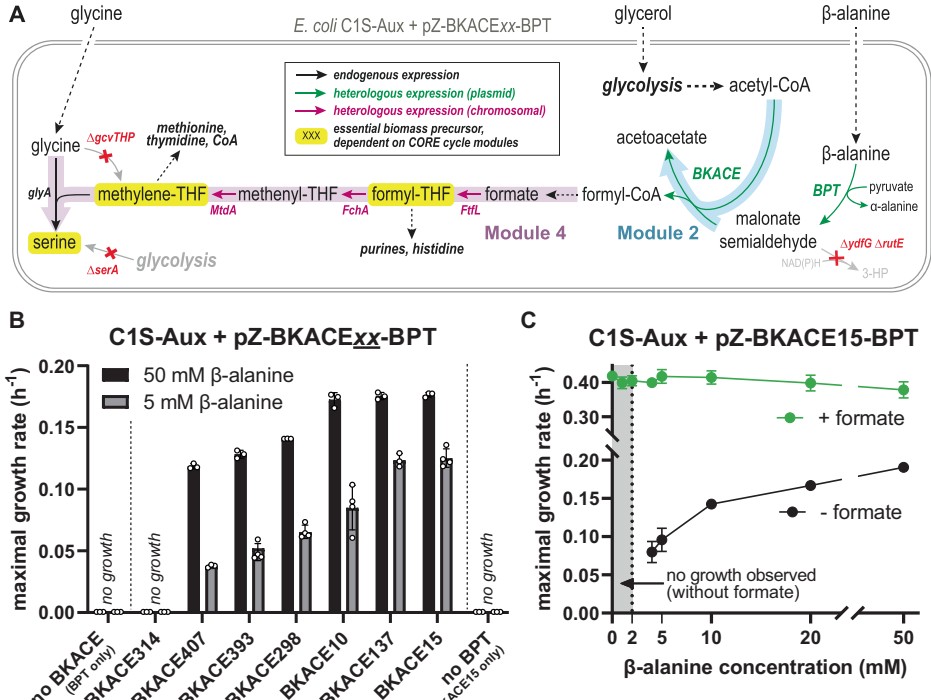

**Fig. 4 | Modules 2 and 4 are functional together in an *E. coli* selection strain.**
**A** Selection scheme to demonstrate CORE cycle functionality by coupling its activity to growth: Rational deletion of *serA* and *gcvTHP* achieves a selection strain ("C1S-Aux") which is auxotrophic to serine and one-carbon units (CORE cycle-dependent biomass precursors highlighted in yellow). Its growth can be rescued by production of these precursors from formate via a heterologous pathway integrated in the chromosome (purple arrows), thus establishing a selection for formate-production via the CORE cycle. Expression of a β-alanine:pyruvate-transaminase (BPT) allows intracellular generation of malonate semialdehyde (MSA), while deletion of two native reductases (*ΔydfG, ΔrutE*) removes sink reactions converting it to 3-hydroxypropionate (3-HP). MSA and acetyl-CoA serve as substrates for BKACE, producing acetoacetate and formyl-CoA, which is hydrolyzed to

formate spontaneously or converted via endogenous thioesterases/CoA-transferases. This metabolic scheme selects for the activity of module 2 and 4 together (blue, pink). **B** When expressed together with BPT, six out of seven tested BKACE-candidates rescue growth of the C1S-Aux strain. No growth is observed for a strain expressing only BPT but lacking BKACE or vice-versa. **C** Growth of the C1S-Aux + pZ-BKACE15-BPT strain is strictly dependent on supplementation of β-alanine in the medium, while the growth rate increases with the supplied β-alanine concentration. Shown data in (**B**) and (**C**) represent mean maximal growth rate values estimated from triplicate cultures (or quadruplicates for selected strains) grown in a 96-well plate. Error bars indicate standard deviation. Source data are provided as a Source Data file.

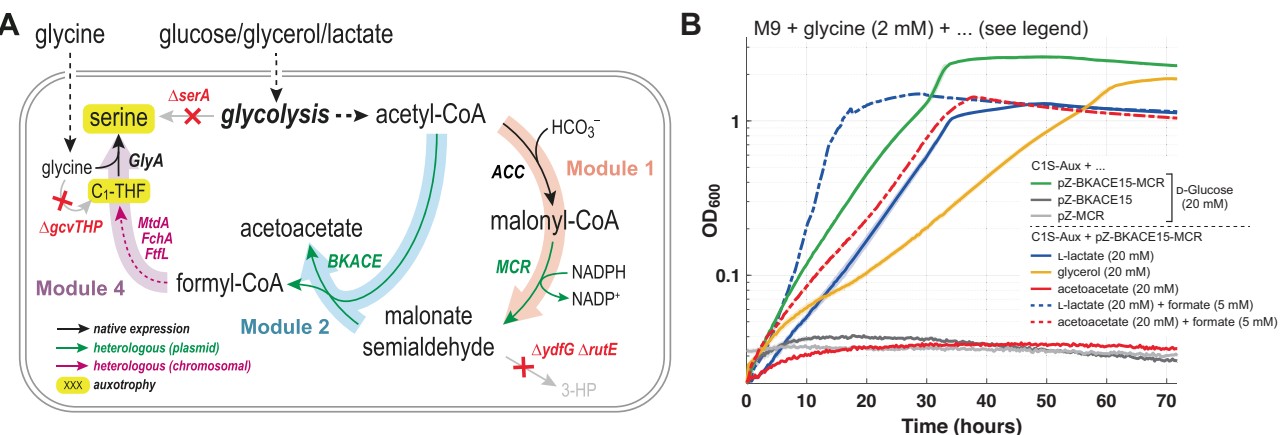

**Fig. 5 | Modules 1, 2 and 4 of the CORE cycle operate robustly together in *E. coli*.**
**A** Selection scheme: Growth of the *E. coli* C1S-Aux strain is used to detect intracellular formate production via the tested modules of the CORE cycle. For details on formate assimilation (purple arrows), see Fig. 4A. By expressing the C-terminal domain of malonyl-CoA reductase from *C. aurantiacus* (MCR) together with BKACE from a plasmid (green arrows), endogenous malonyl-CoA is reduced to MSA, providing the substrate for the BKACE reaction. Malonyl-CoA is replenished by the activity of native *E. coli* acetyl-CoA carboxylase (ACC). **B** The C1S-Aux strain was tested for growth on different carbon sources in M9 minimal medium with

supplementation of glycine (2 mM). Expression of the pathway enzymes (BKACE15 and MCR) rescued growth on different tested carbon sources, such as D-glucose (green line, shortest doubling time ($T_d$) = 4.2 h), L-Lactate (blue, $T_d$ = 5.2 h), or glycerol (yellow, $T_d$ = 8.9 h). In contrast, no growth is observed with acetoacetate (red) as main carbon source, unless formate (5 mM) is supplemented to rescue the auxotrophy (red, dashed). Additionally, no growth is observed in negative control strains lacking MCR or BKACE15, respectively (shown for D-glucose; gray lines). Shown growth curves represent the mean of technical triplicates, with shaded areas indicating the standard deviation. Source data are provided as a Source Data file.

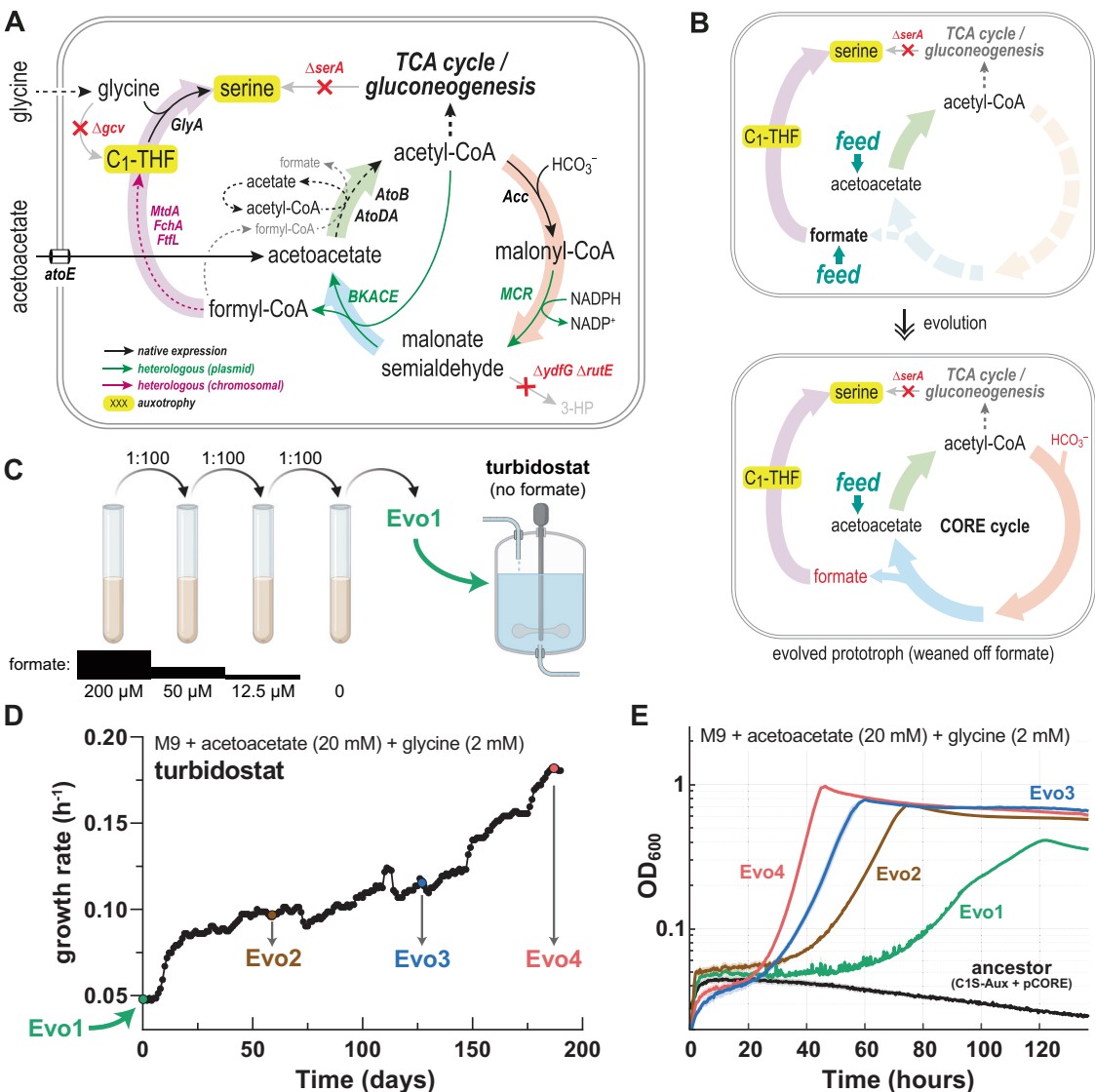

**Fig. 6 | ALE enables operation of the complete CORE cycle. A** Metabolic selection scheme: Feeding acetoacetate as the main carbon source selects for simultaneous operation of all CORE cycle modules 1–4 (see Fig. 3) to enable growth by providing a source of C1-THF and serine. **B** Chosen strategy for adaptive laboratory evolution (ALE). Initial formate-dependent growth on acetoacetate medium (requiring only CORE cycle module 3 and 4) was leveraged as a stepping-stone to wean the selection strain off formate, thereby gradually selecting for additional activity of CORE cycle modules 1 and 2. **C** Formate-independent growth was achieved in a short-term evolution by serial passaging in culture tubes with limiting amounts of supplemented formate. Clone "Evo1" was identified as the best-performing isolate capable of formate-independent growth. Created in BioRender (Satanowski, A.

(2025) https://BioRender.com/u80d598). **D** Long-term evolution in a turbidostat using clone Evo1 as a starting point. Clones were isolated from the population at three indicated time-points. **E** Growth phenotype on acetoacetate medium of isolated clones from three time-points of the turbidostat evolution (Evo2, Evo3, Evo4), compared to the non-evolved ancestral strain (C1S-Aux + pCORE) and the turbidostat starting strain (Evo1). Based on this data, estimated maximal growth rates (GR) and shortest doubling times ($T_d$) are as follows: Ancestor: no growth; Evo1: GR = 0.06 $h^{-1}$; $T_d$ = 11.5 h; Evo2: GR = 0.09 $h^{-1}$, $T_d$ = 7.7 h; Evo3: GR = 0.12 $h^{-1}$; $T_d$ = 5.8 h; Evo4: GR = 0.18 $h^{-1}$; $T_d$ = 3.9 h. Growth curves represent the mean of technical triplicates, with shaded areas indicating the standard deviation. Source data are provided as a Source Data file.

cycle itself, because it is a substrate for the carboxylation reaction and BKACE. We hypothesized that a simple overexpression of enzymes was insufficient to balance fluxes, and thus had prevented growth of the C1S-Aux + pCORE strain on acetoacetate, so far.

Noting that the C1S-Aux + pCORE strain grew when provided with external formate (Fig. 5B), we designed a strategy to gradually evolve growth on acetoacetate with ALE (Fig. 6A, B). We first weaned the strain off its dependence on formate by passaging it via serial dilutions, while gradually decreasing the supplemented formate concentration from 200 μM to 0 μM (see "Methods", Fig. 6C). After four passages (i.e., ~26 generations), one population slowly grew on acetoacetate in the absence of added formate. We isolated three random clones and

confirmed their growth on acetoacetate. The most robustly growing clone, termed "Evo1", reached a maximal growth rate of 0.063 ± 0.001 $h^{-1}$ (corresponding to a doubling time ($T_d$) of 11 h, Supplementary Fig. 20), but still showed much faster growth when formate was added to the medium.

To improve growth of this strain, we leveraged automated ALE. We continuously cultivated Evo1 in a GM3 automated evolution device[48,58,59] for 190 days (6.3 months; i.e., ~750 generations) and repeatedly diluted the culture with fresh acetoacetate medium when it reached an $OD_{600}$ of 0.4, thereby selecting for mutants with an increased growth rate (turbidostat regime). Over the course of this long-term evolution, the apparent growth rate of the population in the

bioreactor improved ~4-fold from 0.046 h$^{-1}$ (doubling time ($T_d$) - 15.1 h) to 0.182 h$^{-1}$ ($T_d$ - 3.8 h), indicating improved operation of the CORE cycle (Fig. 6D).

We isolated a random clone each from three different time-points of the ALE (Evo2, Evo3, Evo4) and compared their growth to the non-evolved "ancestor" (C1S-Aux + pCORE), as well as Evo1, from which ALE was started. As expected, clones Evo2-4 showed increased growth rates compared to Evo1 in acetoacetate medium, while the ancestor remained unable to grow in this condition (Fig. 6E). The fastest clone, Evo4, isolated from the final population, achieved a maximal growth rate of 0.18 h$^{-1}$ on acetoacetate ($T_d$ = 3.9 h; Fig. 6E). Growth on acetoacetate was abolished upon curing the pCORE plasmid from clones Evo1-4, demonstrating that the heterologous CORE cycle enzymes were essential (Supplementary Fig. 21, further discussed below). To confirm that Evo3 and Evo4 were indeed using the CORE cycle, we used 1,3-$^{13}$C$_2$-labeled acetoacetate as carbon source, which resulted in a serine and methionine labeling pattern that is consistent with a fully labeled C1-THF pool (Supplementary Fig. 22). Altogether, this data validated operation of the complete CORE cycle in the evolved *E. coli* strains.

### Genetic basis of evolved growth via the CORE cycle

To understand the genetic adaptations along the ALE trajectory, we re-sequenced the genome and the plasmids of the non-evolved ancestor C1S-Aux and clones Evo1-4. Evo1, 2 and 3 contained nine distinct mutations in total, while Evo4 had accumulated 81 mutations (Supplementary Table 6), likely due to a transposon-mediated disruption of the *mutT* gene, which is known to increase the global mutation rate by more than thousandfold[52,60–62]. Emergence of such hypermutator genotypes has been observed previously in ALE experiments, where it often serves as an "enabler" mutation[63–66].

To trace the appearance of this hypermutation genotype in our ALE, we isolated and sequenced additional random clones from the same Evo1-4 populations. This showed that the *mutT* mutation must have appeared between Evo2 and Evo3. Almost all clones of the Evo3 population already carried this mutation, which had completely taken over the Evo4 population (for all sequencing results, see Supplementary Data 1). Overall, we discovered 252 distinct mutations among the various clones, but decided to focus on two that were conserved in all clones and had affected two enzymes of the CORE cycle, MCR and AtoA.

Despite sequencing single clones, we found mixed sequencing reads for the MCR gene (encoded on the pCORE plasmid), indicating presence of both a wild-type and a mutated (i.e., truncated) MCR Y331* variant (Supplementary Table 6, Supplementary Data 1). We hypothesized that the mixed sequencing reads were caused by multimerized plasmids (concatemers)[67–69]. We isolated the pCORE plasmids from Evo1-4 and confirmed the presence of concatemers in all evolved clones (Supplementary Fig. 23). Cell-extract assays showed that MCR activity was reduced between ~60–90% in clones carrying the concatemers (Supplementary Fig. 24). However, we found that concatemer formation was helpful, but not necessary for growth on acetoacetate, as follows: When transforming the non-evolved C1S-Aux with the isolated concatemer, this did not restore growth on acetoacetate. In contrast, when (re-)transforming a cured Evo3 strain with either the original pCORE plasmid, or a plasmid in which MCR expression was driven from a 10-fold weaker RBS[41,70] ("pCORE$_{rbsC}$"), or the concatemer ("pCORE$_{Evo3}$"), we observed growth in all cases. Yet, Evo3 strains transformed with pCORE$_{rbsC}$ grew slightly better, indicating that a reduced MCR activity was helpful to operate the CORE cycle in vivo (Supplementary Fig. 21A).

In case of *atoA*, the gene had independently mutated (at least twice) during ALE, resulting in an AtoA I25L mutation in Evo1 and all derived clones (i.e., Evo2-4), while a parallel lineage ("Evo1a") had acquired an AtoA A65S mutation (Supplementary Data 1). Homology

modelling located I25 and A65 in the active site facing the acyl-moiety, indicating that these mutations might have altered substrate specificity or activity (Supplementary Fig. 25A). We tested the effect of these two mutations and the double variant (AtoA I25L A65S) on acetoacetate activation with acetyl-CoA and formyl-CoA (Supplementary Figs. 25B–D, Supplementary Table 7). While all variants exhibited lower activity compared to the WT, AtoA I25L (and the double variant) showed a 3- to 5-fold increased relative activity with formyl-CoA compared to acetyl-CoA, indicating that these variants had acquired the capability to activate acetoacetate more preferentially with formyl-CoA, thus integrating better with the CORE cycle (Fig. 6A).

### The CORE cycle could enable synthetic autotrophy and enhance photosynthesis

Having demonstrated a functional CORE cycle in vivo, we finally analyzed its potential in improving or replacing natural CO$_2$ fixation (Fig. 1C, D). Using flux balance analysis[71], we first evaluated the CORE cycle in combination with the Calvin-Benson-Bassham (CBB) cycle of photosynthesis. The CORE cycle can replace (or supplement) natural photorespiration[22,72–75] and circumvent the wasteful decarboxylation of glycine by providing an alternative route to methylene-THF (Fig. 1D, Supplementary Fig. 26). This converts photorespiration into a process that assimilates CO$_2$ rather than releasing it, thus doubling its carbon efficiency from 75% to 150%[22,33,75,76]. In terms of energetic efficiency, the CORE cycle requires 27% less ATP and 16% fewer reducing equivalents compared to native photorespiration and outperforms it by 25% in terms of 3PG yield (Supplementary Fig. 27, Supplementary Table 8).

We also assessed the CORE cycle as a CO$_2$-fixation pathway for synthetic autotrophy[22,77–81], for which we combined the CORE cycle with different downstream formate-assimilation pathways[82]. We modelled CO$_2$-fixation efficiency of the CORE cycle in combination with the serine cycle[42,44,83], the reductive glycine pathway[43], the formolase pathway[84,85], and the ribulose monophosphate (RuMP) cycle[86,87]. This showed that all CORE cycle-based pathways outperformed the CBB cycle in terms of biomass yield (when considering RuBisCO oxygenation, Supplementary Fig. 28). Overall, these results showcase the flexibility of an oxygen-tolerant reduction-first strategy to support or replace natural CO$_2$ fixation pathways.

## Discussion

In this work, we developed and realized the CORE cycle – an artificial, modular metabolic pathway for the conversion of CO$_2$ to formate. The CORE cycle allows CO$_2$-reduction at fully aerobic conditions and atmospheric CO$_2$ levels with NADPH as a reductant. We show that this metabolic route provides sufficient flux to satisfy the cellular demand for C1-units and serine in engineered *E. coli* strains, enabling a doubling time of roughly 4 h, which we achieved by a combination of rational metabolic engineering and ALE.

When coupled to downstream formate-assimilation pathways, the CORE cycle opens up new possibilities for CO$_2$-based bioproduction (Supplementary Fig. 28). While we demonstrate the successful production of cellular C1-units from CO$_2$ in this study, future efforts could open the way to more complex compounds, especially, when the CORE cycle is used in combination with recently developed synthetic pathways, such as the ATP-driven reduction of formate to formaldehyde[84,88] together with the homoserine cycle[89], the EuMP cycle[90], the FORCE pathway[91–93], or formolase[85].

Used as a CO$_2$-assimilating photorespiratory bypass, the CORE cycle carries the potential to enhance natural carbon fixation by providing an intracellular formate pool from CO$_2$, which could increase photosynthetic yield by 25% (Supplementary Fig. 27). In fact, it is tempting to speculate that such a "formate-assisted photorespiration" may occur in natural autotrophic, aerobic organisms which operate the CBB cycle in presence of reduced C1-compounds (e.g. methane, methanol, formate). In such organisms, these C1-compounds could

serve not only as an energy source[94–96], but also as a source of C1-THF to support phosphoglycolate salvage[97], although this remains to be demonstrated.

We note that recently, the synthetic 4-hydroxy-2-oxobutanoic acid (HOB) cycle was presented, which in principle could also be considered as a $CO_2$-reduction pathway[98]. The HOB cycle converts $CO_2$ into a THF-bound formaldehyde-moiety (methylene-THF) but has yet to be fully demonstrated in vivo (i.e., as a complete cycle). More importantly, this pathway does not produce a free C1 molecule, such as formate in case of the CORE cycle, making the HOB cycle less flexible in terms of downstream assimilation strategies (see above).

Overall, our study expands the natural solution space of "reduction-first"-based $CO_2$-fixation by an oxygen-tolerant alternative, opening up this highly efficient strategy for aerobic organisms, which provides multiple opportunities to enhance or replace natural and synthetic $CO_2$ fixation in the future.

## Methods

### Synthesis of chemical compounds

Malonate semialdehyde (MSA) was synthesized chemically[99], as follows: Alkaline hydrolysis of ethyl 3,3-diethoxypropionate (EDEOP, Sigma Aldrich) was followed by acid hydrolysis to yield MSA. 0.5 mL EDEOP were added to 7.5 mL of 0.5 M NaOH and incubated for 1 h at RT. Afterwards 5 mL 10% HCl were added and incubated for 1 h at RT. The solution was cooled on ice, 13 mL 1 M $K_xH_yPO_4$ buffer pH 6 were added and the pH was adjusted to pH 6. The solution was stored at −80 °C. The yield of MSA was determined enzymatically with full-length MCR from *C. aurantiacus* or 3-hydroxybutyrate dehydrogenase. The synthesis yield of MSA was estimated to be over 98%. Due to its instability, MSA was prepared freshly for each quantitative measurement.

Formyl-CoA was synthesized as follows, using slight modifications of established methods[100–102]. Initially, formyl thiophenol was prepared by adding 5.8 mL (150 mmol, 3 eq.) of formic acid dropwise to 7.1 mL (75 mmol, 1.5 eq.) of acetic anhydride, followed by stirring at 25 °C for 2.5 h. Next, 61 μL (0.75 mmol, 0.015 eq.) of pyridine and 5.1 mL (50 mmol, 1 eq.) of thiophenol were added, and the mixture was stirred overnight. Impurities were removed using a rotary evaporator at 50 °C and 25 mbar. The remaining mixture was washed with cold brine, dried over $MgSO_4$, and distilled at 131 °C and 50–60 mbar to yield a clear oil, which was stored under nitrogen at −20 °C. Next, to synthesize formyl-CoA, 200 mg of CoA were dissolved in 2 mL of ice-cold 1 M $KHCO_3$ at pH 8.0 and degassed by shaking. Then, 0.4 mL of formyl thiophenol was added, and the mixture was shaken vigorously for 10 min. Cold diethyl ether was used to wash the product, removing phenol and formic acid. The pH was then lowered to below 4 by adding HCl, followed by two additional ether washes. The reaction product was purified by HPLC-MS[101].

Acetyl-CoA was synthesized by symmetric anhydride synthesis[103]. 200 mg CoA (0.25 mmol, 1 eq.) was mixed with 1.6 eq. of acetic anhydride (45 μL, 0.41 mmol) in 5 mL 0.5 M $NaHCO_3$ and stirred on ice for 45 min. Next, the pH of the solution was lowered to below 4 by adding formic acid and the reaction product was purified by HPLC-MS[103].

1,3-$^{13}C_2$-acetoacetate was synthesized by alkaline hydrolysis of commercially available 1,3-$^{13}C_2$-ethyl acetoacetate (Scientific Laboratory Supplies, Cat. No. 485640). A 1.3:1 ratio of NaOH and 1,3-$^{13}C_2$-ethyl acetoacetate was mixed by dissolving 208 mg NaOH in 2 mL $H_2O$ and adding the labeled compound. After 48 h incubation at room temperature and stirring conditions, the sample was frozen in liquid nitrogen, lyophilized and its concentration and purity validated by LC-MS analysis.

### Protein production and purification

The collection of β-keto acid cleavage enzymes (BKACE) considered here was previously described[29]. Briefly, the forward primers introduced a hexahistidine sequence in the proteins after the initial methionine for purification purposes[104]. For each construction, the plasmid was transformed into BL21-CodonPlus (DE3)-RIPL cells.

For the production of cell lysates to screen the whole BKACE collection, corresponding plasmids were transformed into *E. coli* BL21-CodonPlus (DE3)-RIPL cells and grown in 96-well in 1.6 mL of Terrific Broth medium containing 0.5 M Sorbitol, 5 mM betaine and 100 μg/mL carbenicillin for 4.5 h at 37 °C. Isopropyl β-D-thiogalactopyranoside (IPTG) was added at a concentration of 500 μM to induce protein production, and the cells were further grown at 20 °C overnight. The cells were harvested by centrifugation and suspended in 300 μL of 50 mM Tris-HCl, pH 7.5 containing 10% glycerol, 1 mM Pefabloc SC and 0.2 μL of Lysonase™ Bioprocessing reagent (Merck NovagenR), and sonicated using a Branson 2510 sonication water bath. After centrifugation, the clarified lysate was analyzed by SDS-PAGE to check for recombinant protein production and stored at −80 °C. Protein concentration was determined by the Bradford assay with Bovine serum albumin as standard (Bio-Rad).

For the purification of BKACE homologs for in-depth characterization, the corresponding plasmid was transformed into chemically competent *E. coli* Arctic Express DE3. Cells were grown on lysogeny broth (Miller recipe) agar plates containing 34 μg/mL chloramphenicol an incubated over night at 37 °C. 2 L Terrific Broth medium (TB; 24 g/L yeast extract, 12 g/L tryptone, 4 mL/L glycerol, 17 mM $KH_2PO_4$, 72 mM $K_2HPO_4$) was inoculated from the agar plate and incubated at 37 °C and 140 rpm. At $OD_{600} = 0.4–0.6$ protein expression was induced with 500 μM IPTG and cells were incubated over night at 16 °C. Cell harvesting at 8000 g and 4 °C for 12 min, lysis by French pressing at 18,000 psi and ultracentrifugation at 100,000 g was followed by His-Trap purification using an Äkta Start (GE Healthcare) linked to a HisTrap FF column (GE Healthcare). The purification buffer contained 50 mM HEPES pH 7.8, 500 mM KCl, 1 M proline and 1 mM $ZnCl_2$. The elution was done with 500 mM imidazole. Protein desalting occurred via gel-filtration chromatography using a HiLoad 16/600 Superdex 200 pg column (GE Healthcare) and a buffer containing 50 mM HEPES pH 7.8., 150 mM KCl, 1 M proline and 1 mM $ZnCl_2$. Desalting of protein samples for ICP-OES measurements was done by gel-filtration using the same buffer but without $ZnCl_2$. Protein quantification occurred by absorbance measurement at 280 nm. Protein purity was validated by SDS-PAGE using 15 μg of purified protein on a 4–20% Mini-Protean TGX Precast Protein Gel (Biorad).

Protein production and purification of Pcc_Me_D407I was done using an *E. coli* BL21 DE3 strain that contains a biotin ligase (BirA) from *Methylobacterium extorquens* and 6 L of TB containing additionally 2 μg/mL biotin. The purification buffer contained 50 mM HEPES pH 7.8 and 500 mM KCl and the desalting buffer contained 50 mM HEPES pH 7.8 and 150 mM KCl. All other steps remained the same as above.

For the production of all other proteins, the corresponding plasmid was transformed into chemically competent *E. coli* BL21 DE3. Cells were grown on lysogeny broth (Miller recipe) agar plates with appropriate antibiotics an incubated over night at 37 °C. 1 L TB medium was inoculated from the agar plate and incubated at 37 °C and 140 rpm. At $OD_{600} = 0.4–0.6$ protein expression was induced with 500 μM IPTG and cells were incubated over night at 25 °C. Cell harvesting at 8000 g and 4 °C for 12 min, cell lysis using a Sonopuls GM200 sonicator (BANDELIN Electronic) at an amplitude of 50% with five consecutive cycles of 30 pulses over 60 s and ultracentrifugation at 100,000 g was followed by His-Trap purification using an Äkta Start (GE Healthcare) linked to a HisTrap FF column (GE Healthcare). The purification buffer contained 50 mM HEPES pH 7.8 and 500 mM KCl. The elution was done with 500 mM imidazole. Protein desalting occurred using a HiTrap Desalting column (GE Healthcare) and a buffer containing 50 mM HEPES pH 7.8. and 150 mM KCl. Protein quantification and purity validation occurred as described above.

## Spectrophotometric assays

To measure NAD(P)H consumption or production over time, spectrophotometric assays with the Agilent Technologies Cary 60 UV-Vis device were done. All assays were carried out in a reaction volume of 200 µL using a quartz cuvette with 10 mm light path (Hellma Analytics) and the Kinetics software from Agilent.

To measure acetyl-CoA carboxylase (Pcc_Me_D407I) activity, a coupled enzyme assay with the full-length malonyl-CoA reductase (Mcr_Ca) from *Chloroflexus aurantiacus* was performed[105], as follows: 100 mM MOPS/KOH pH 7.8, 50 mM KHCO$_3$, 2 mM ATP, 0.3 mM NADPH, 5 mM MgCl$_2$, 2.8 mg/mL Mcr_Ca and 0.01–1 mg/mL Pcc_Me_D407I were mixed in a cuvette and incubated for 2 min at 37 °C. The reaction was started with 1 mM acetyl-CoA and absorption was measured over time at 340 nm.

To measure acetoacetate-CoA ligase (Acl_Sz/Acl_Sl) activity, a coupled enzyme assay with 3-hydroxybutyryl-CoA dehydrogenase (PhaB) from *Ralstonia eutropha* was performed. 100 mM MOPS/KOH pH 7.8, 0.6 mM NADH, 2.5 mM ATP, 2 mM acetoacetate, 5 mM MgCl$_2$, 0.075 mg/mL PhaB and 0.001–0.05 mg/mL Acl_Sz/Acl_Sl were mixed in a cuvette and incubated for 2 min at 37 °C. The reaction was started with 1 mM CoA and absorption was measured over time at 340 nm.

To measure malonyl-CoA reductase activity of purified proteins, 100 mM MOPS/KOH pH 7.8, 0.3 mM NADPH, and 0.08–0.32 mg/mL MCR were mixed in a cuvette and incubated for 2 min at 20 - 37 °C. The reaction was started with 1 mM malonyl-CoA and absorbance was measured over time at 340 nm. MCR of *Sulfolobus tokodaii* (Mcr_St) and the dissected C-terminal domain of *Chloroflexus aurantiacus* (MCR) were used. To measure malonyl-CoA reductase activity of clarified *E. coli* cell lysate, 0.04–1.7 mg/mL total proteins were used.

To measure β-ketothiolase (AtoB) activity, a coupled enzyme assay with propionaldehyde dehydrogenase (PduP) from *Rhodopseudomonas palustris* was performed. Acetoacetyl-CoA was produced in situ with Acetoacetate:CoA ligase (Acl_Sz) from *Shinella zoogloeoides*. 100 mM MOPS/KOH pH 7.8, 0.3 mM NADH, 2.5 mM ATP, 5 mM MgCl$_2$, 5 mM acetoacetate, 0.1 mg/mL Acl_Sz, 0.05 mg/mL PduP and 1.5–15 µg/mL AtoB were mixed in a cuvette and incubated for 2 min at 37 °C. The reaction was started with 2 mM CoA and absorbance was measured over time at 340 nm.

To measure the metal dependence of β-keto acid cleavage enzyme (BKACE15), a coupled enzyme assay was performed with malic dehydrogenase from porcine heart (MDH, Merck, CAS number 9001-64-3) and citrate synthase from porcine heart (CS, Merck, CAS number 9027-96-7). Oxaloacetate production by MDH is thermodynamically hindered and only occurs when oxaloacetate is drained by CS, requiring the co-substrate acetyl-CoA that is produced by the (reverse) BKACE reaction. 100 mM TRIS/HCl pH 7.5, 1 mM NAD$^+$, 10 mM malate, 20 mM acetoacetate, 27.5 units MDH, 13.5 units CS, 0.1–0.5 mg/mL BKACE15, were mixed in a cuvette with either 5 mM ZnCl$_2$, 5 mM MgCl$_2$, 5 mM MnCl$_2$ or no metals and incubated for 2 min at 37 °C. The reaction was started with 0.5 mM formyl-CoA and absorbance was measured over time at 340 nm. BKACE15 activity was observed only upon (re-)addition of ZnCl$_2$.

## LC-MS-based assays for BKACE activities

The β-keto acid cleavage enzyme (BKACE) collection screen was performed in recombinant cell-free extracts in 96 well-microplates at room temperature for 2 h. For each candidate enzyme roughly 12 µg of protein were incubated in 100 µL of 50 mM Tris-HCl (pH 7.5) containing 250 µM acetyl-CoA, 1.2 mM MSA. Due to the instability of acetoacetate in the ion source that prevents its reliable detection and quantitation, it was chemically derivatized with 2-Hydrazinoquinoline (2-HQ), to enhance chromatographic performance, stability and detectability by LC-MS[106]. Specifically, product formation was tracked by detecting the loss of CO$_2$ from derivatized acetoacetate after CID fragmentation. 5 µL of the reaction were mixed with 100 µL of acetonitrile containing 50 µM 2-HQ and incubated for 1 h on ice. The derivatized sample was finally filtered on an AcroprepAdv Multi-well filter plate 0.2 µM WWPTFE (PALL). Experiments were conducted in duplicates.

For all BKACEs that reacted with MSA, proteins were purified and activities were quantified as follows: 3 µg of enzyme were incubated in 100 µL of 50 mM Tris-HCl (pH 7.5) containing 250 µM acetyl-CoA, 1.2 mM MSA. At different times (0, 30, 60 and 120 min) 5 µL of the reaction mixture were mixed with 100 µL of acetonitrile containing 50 µM 2-HQ and incubated for 1 h on ice. The derivatized sample was finally filtered on an AcroprepAdv Multi-well filter plate 0.2 µM WWPTFE (PALL). Experiments were conducted in duplicates. Accumulation of acetoacetate over time was linear during this period of time. Acetoacetate was quantified using a derivatized standard acetoacetate calibration range.

LC-MS analyses of the samples were conducted using a Dionex UltiMate TCC-3000RS chromatographic system (Thermo Fisher Scientific) coupled to a Velos Pro Orbitrap Elite mass spectrometer (Thermo Fisher Scientific) fitted with a heated electrospray ionization source (HESI) operating in the positive ionization mode. The ionization spray (IS) was set to +3.5 kV and the capillary temperature at 275 °C. Sheath gas and auxiliary gas flow rates were set at 60% arbitrary units (a.u.) and 44 a.u., respectively. The S-lens RF level was set to 55%. Data were acquired in raw files and processed with the Qualbrowser module of Xcalibur 2.2 (Thermo Fisher Scientific) to access to elemental compositions. Fragmentation experiments were performed under collision induced dissociation (CID) with a normalized collision energy (NCE) of 22%. Mass spectra were acquired over an *m/z* range from *m/z* 50 up to *m/z* 1000 with the mass resolution set to 60,000 FWHM at *m/z* 400.

Chromatographic separation was achieved on an XBridge C18 column (150 × 4.6 mm; 5 µm; Waters) thermostated at 40 °C. The mobile phase flow rate was set at 0.3 mL/min, and 10 µL were injected. Mobile phase A consisted of 0.05% aqueous acetic acid containing 2 mM ammonium acetate, and mobile phase B consisted of 95% aqueous acetonitrile containing 0.05% acetic acid and 2 mM ammonium acetate. The gradient started at 100% A for 2 min, followed by a linear gradient to 100% B for 10 min, and remained at 100% B for 10 min. The system returned to the initial solvent composition in 2 min and re-equilibrated under these conditions for 3 min.

## In-depth kinetic characterization of BKACE candidates

Enzyme activity assays for in-depth characterization of selected BKACE candidates from the collection screen were conducted in an optimized assay with purified proteins. A reaction mix containing 100 mM MOPS/KOH pH 7.8, 4 mM MSA, 1 mM ZnCl$_2$ and 0.1 mg/mL BKACE was prepared in a volume of 160 µL and incubated at 37 °C for 1 min. The reaction was started by adding 1 mM acetyl-CoA and samples were quenched after 30, 60, 90 and 120 s in 10% formic acid. Quenched samples were analyzed by quantifying acetoacetate, formyl-CoA and acetyl-CoA via LC-MS as described below. Michaelis-Menten kinetics for MSA were conducted using the same assay and 0.5–50 mM MSA. Michaelis-Menten kinetics for acetyl-CoA were conducted using the same assay and 0.025–2 mM acetyl-CoA. Product inhibition by acetoacetate to determine IC$_{50}$ was measured using the same assay and 5 mM MSA, 1 mM acetyl-CoA and 0–20 mM acetoacetate. Relative activities were calculated by normalizing the data to the activity with 0 mM acetoacetate. Product inhibition by acetoacetate to investigate the type of inhibition was measured using the same LC-MS assay and formyl-CoA for detection. 2 mM acetyl-CoA, 0.5–4 mM MSA and 0–8 mM acetoacetate were used.

Quantitative determination of the acetyl-CoA, formyl-CoA and CoA was performed using a LC-MS/MS. The chromatographic separation was performed on an Agilent Infinity II 1290 HPLC system using a Kinetex EVO C18 column (150 × 2.1 mm, 3 µm particle size, 100 Å pore

size, Phenomenex) connected to a guard column of similar specificity (20 × 2.1 mm, 3 μm particle size, Phenomoenex) at a constant flow rate of 0.25 mL/min with mobile phase A being 50 mM Ammonium Acetate in water at a pH of 8.1 and phase B being 100% methanol (Honeywell, Morristown, New Jersey, USA) at 40 °C. The injection volume was 2 μL. The mobile phase profile consisted of the following steps and linear gradients: 0–1 min constant at 2.5% B; 1–6 min from 2.5 to 95% B; 6–8 min constant at 95% B; 8–8.1 min from 95 to 2.5% B; 8.1 to 10 min constant at 2.5% B. An Agilent 6470 mass spectrometer was used in positive mode with an electrospray ionization source and the following conditions: ESI spray voltage 4500 V, nozzle voltage 1500 V, sheath gas 400 °C at 11 L/min, nebulizer pressure 30 psi and drying gas 250 °C at 11 L/min. Compounds were identified based on their mass transition and retention time compared to standards. Chromatograms were integrated using MassHunter software (Agilent, Santa Clara, CA, USA). Relative abundance was determined based on the peak area. Mass transitions, collision energies, Cell accelerator voltages and Dwell times have been optimized using chemically pure standards. Parameter settings of all targets are given in Supplementary Table 9.

Quantitative determination of Acetoacetate was performed using LC-MS/MS. The chromatographic separation was performed on an Agilent Infinity II 1290 HPLC system using a Kinetex EVO C18 column (150 × 2.1 mm, 3 μm particle size, 100 Å pore size, Phenomenex) connected to a guard column of similar specificity (20 × 2.1 mm, 3 μm particle size, Phenomoenex) at a constant flow rate of 0.2 mL/min with mobile phase A being 0.1% formic acid in water and phase B being 0.1% formic acid in methanol (Honeywell, Morristown, New Jersey, USA) at 25 °C. The injection volume was 0.5 μL. The profile of the mobile phase consisted of the following steps and linear gradients: 0–5 min constant at 0% B; 5–6 min from 0% to 100% B; 6–8 min constant at 100% B; 8–8.1 min from 100% to 0% B; 8.1–12 min constant at 0% B. An Agilent 6470 mass spectrometer was used in negative mode with an electrospray ionization source and the following conditions: ESI spray voltage 2000 V, nozzle voltage 500 V, sheath gas 300 °C at 11 L/min, nebulizer pressure 45 psi and drying gas 170 °C at 5 L/min. Compounds were identified based on their mass transition and retention time compared to standards. Chromatograms were integrated using MassHunter software (Agilent, Santa Clara, CA, USA). Absolute concentrations were determined based on an external Standard curve. Mass transitions, collision energies, Cell accelerator voltages and Dwell times have been optimized using chemically pure standards. Parameter settings of all targets are given in Supplementary Table 10.

### Crystallization and structure determination of BKACE15

The sitting-drop vapor-diffusion method was used for crystallization at 16 °C. For the crystallization of the apo-enzyme purified BKACE15 (8.6 mg/mL) in 50 mM HEPES pH 7.8, 150 mM KCl, 1 M L-proline, and 1 mM $ZnCl_2$ was mixed in a 1:1 ratio with 25% (w/v) pentaerythritol propoxylate (17/8 PO/OH), 100 mM HEPES pH 7.5. The final size of the drops was 1 microliter. Prior to flash freezing the crystals in liquid nitrogen, the mother liquor was supplemented with 37% (w/v) pentaerythrol propoxylate (17/8 PO/OH).

Co-crystallization in the presence of malonate was performed by mixing BKACE15 (8.6 mg/mL) in 50 mM HEPES pH 7.8, 150 mM KCl, 1 M proline, and 1 mM $ZnCl_2$ in a 1:1 ratio with 20% w/v PEG3350, 200 mM di-sodium malonate pH 7.0. The final size of the drops was 1 μL. Prior to flash freezing the crystal in liquid nitrogen, the mother liquor was supplemented with 10 mM CoA and 40% (w/v) PEG200.

To acquire a crystal structure with bound acetyl-CoA and acetoacetate BKACE15 (8.6 mg/mL) in 50 mM HEPES pH 7.8, 150 mM KCl, 1 M L-proline, and 1 mM $ZnCl_2$ was mixed in a 1:1 ratio with 10% (v/v) pentaerythritol ethoxylate, 10% v/v 1-Butanol. The final size of the drops was 1 microliter. Prior to flash freezing the crystals in liquid nitrogen, the mother liquor was supplemented with 15 mM acetyl-CoA, 20 mM acetoacetate, and 20% (w/v) pentaerythrol propoxylate (17/8 PO/OH).

X-ray diffraction data (Supplementary Table 11) were collected at beamlines PETRA III P13 and P14 of DESY (Deutsches Elektronen-Synchrotron, Hamburg). Data were processed with the XDS[107] and CCP4 v.8.0 software suite[108]. Structures were solved by molecular replacement using Phaser of the Phenix software package (v1.20.1)[109], built with Phenix.Autobuild and refined with Phenix.Refine. Additional modelling, manual refinement, and ligand fitting was done in Coot (v.0.9.8.3)[110]. Final positional and B-factor refinements, as well as water picking, were performed using Phenix.Refine. The metal dependence of BKACE was tested spectrophotometrically as described above. Structural models for the apo-enzyme, enzyme with bound CoA and malonate, as well as enzyme with bound acetyl-CoA and acetoacetate were deposited to the Protein Data Bank in Europe (PDBe) under PDB accession 8RIO, 8RIP, and 9HNF, respectively (Supplementary Table 11). Figures were made using PyMOL Molecular Graphics System (version 2.5.7; Schrödinger).

### Metal analysis

Inductively coupled plasma optical emission spectroscopy (ICP-OES) was used to identify metals bound to BKACE15. For sample preparation, 236 μg of BKACE15 were dissolved in 0.5 mL of trace metal grade concentrated nitric acid and incubated for 12 h at 25 °C. Subsequently, the samples were boiled for 2 h at 70 °C before they were diluted 17-fold in distilled water. The metal content was analyzed with a 720/725 ICP-OES device (Agilent Technologies) and a λ = 213.857 nm for zinc. The device was operated with ICP Expert v4.1.0 software (Agilent Technologies). Zinc was quantified using ICP multi-element standard solution IV (Merck) as a standard. The results were plotted using GraphPad Prism v10.

### Structural modelling and analysis

Structural investigation of *atoA* mutations was performed via homology modelling using SWISS-MODEL[111]. As a template, the structure of succinyl-CoA:3-ketoacid CoA transferase from pig heart (PDB 3OXO, 45% amino acid identity) was used, which had a CoA bound that was covalently linked to an active site glutamate.

### Strains and genomic modifications

All strains used in this study are listed in Supplementary Table 12. All selection strains are based on *E. coli* SIJ488[112], a derivative of *E. coli* MG1655. The SIJ488 strain is equipped with inducible genes for λ-Red recombineering (Red recombinase system; flippase) integrated into its chromosome to increase ease-of-use for multiple genomic modifications. Using a published strain as a starting point (Supplementary Table 12), the additional gene deletions were performed by λ-Red recombineering[113], using linear dsDNA fragments produced by PCR of an antibiotic resistance cassette ($Cm^R$ or $Km^R$) with primers carrying 50 bp overhangs serving as homologous regions flanking the target gene (for oligonucleotides, see Supplementary Table 13). For deletion of *rutE*, the $Cm^R$ resistance cassette from pKD3[113] was amplified. For deletion of *ydfG*, the $Km^R$ resistance was amplified from the template "FRT-PGK-gb2-neo-FRT" (Cat. No. A002, Gene Bridges GmbH, Germany). The PCR product (400 ng) was electroporated into the desired strain, using cells harvested from a mid-log phase culture in which recombination proteins had been induced with 15 mM L-Arabinose for 45 min at $OD_{600}$ ~ 0.3. Strains were verified as described below.

For chromosomal integration of the heterologous genes (required for CORE cycle module 4, i.e. formate assimilation/C1-THF metabolism), we used a previously described genome insertion protocol[41] and a published construct containing the desired genes ("pDM4:SS9-$C_1$M")[39]. In brief, a non-replicative plasmid (pDM4, R6K ori) was introduced into the recipient strain via conjugation from an *E. coli* ST18 donor strain. Antibiotic resistance was used to select for chromosomal insertion via native homologous recombination based on 600 bp homology regions, with subsequent levansucrase (*sacB*)

counter-selection. Serendipitously, a clone was identified in which the complete operon had inserted not in the previously used genomic locus ("SS9"[39,114]), but instead, in a non-coding region in-between *nhaR* and *rpsT*, likely due to unintended homology of this region with the pDM4 backbone. The insertion site was confirmed by PCR, Sanger sequencing and short-read sequencing. This clone was used for all further steps.

Strains with the desired deletions or chromosomal insertion were selected by plating on appropriate antibiotics (kanamycin or chloramphenicol) and confirmed by determining the size of the respective genomic locus via PCR (oligonucleotide sequences are shown in Supplementary Table 13). For removal of resistance markers flanked by FRT sites, flippase was induced in growing cultures at $OD_{600}$ ~ 0.2 by adding 50 mM L-rhamnose, followed by cultivating at 30 °C for ~4 h. Loss of the antibiotic resistance was confirmed by identifying individual colonies that required absence of the respective antibiotic, and by PCR of the genomic locus (oligonucleotide sequences are shown in Supplementary Table 13). Plasmids were transformed into selection strains by electroporation. Presence of the plasmid was verified by antibiotic resistance and PCR.

### Cloning of expression constructs for protein purification
All plasmids used in this study are listed in Supplementary Data 2. Cloning of expression constructs for Acl_Sl, Acl_Sz, MCR and all BKACE-variants (for protein purification) was done using modular cloning and the "Marburg Collection"[115]. Target genes were either synthesized or PCR amplified from the host's genome using Q5 polymerase from NEB and the dedicated protocol. Golden gate cloning was followed by DpnI digestion, transformation into *E. coli* NEB Turbo and colony picking to yield the correct construct that was validated by sequencing.

Cloning of the expression construct for AtoDA was done by Golden gate cloning of synthesized genes into pET28a, followed by DpnI digestion, transformation into *E. coli* NEB Turbo and colony picking to yield the correct construct that was validated by sequencing.

Introduction of point mutations to construct mutant variants of BKACE15 and AtoA was done by oligonucleotide-based site-directed mutagenesis[116]. pME_B_0_04_0055_BKACE15 and pTE3156 were used as templates for mutagenesis. The PCR was followed by DpnI digest and a PCR clean-up. The cloning mixture was transformed into *E. coli* NEB Turbo and colonies picked for sequence validation.

For transformation of cloned constructs, various amounts of DNA were added to 50 μL of chemically competent *E. coli* DH5α or *E. coli* NEB Turbo and incubated on ice for 30 min. A heat shock at 42 °C for 45 s was followed by 10 min of incubation on ice. 600 μL S.O.C. medium (Invitrogen) was added, and cells were incubated for 1 h at 37 °C while shaking at 200 rpm. Finally, cells were transformed on LB plates with selection marker.

### Plasmid construction for in vivo testing of CORE cycle modules
Cloning was performed in *E. coli* DH5α. All plasmids used in this study are listed in Supplementary Data 2. Heterologous genes were codon-optimized for *E. coli* K−12 and synthesized by TWIST Biosciences. All heterologous genes for in vivo testing were expressed with an N-terminal 6xHisTag without a linker, to provide a consistent sequence immediately downstream of the ribosomal binding site (RBS), aiming to reduce sequence-dependent differences in translation initiation rate[70].

Genes were cloned into a pZASS backbone (p15A ori, Strep[R])[41] containing a constitutive promoter ("pgi#20")[117]. A medium-strength ribosome binding site (RBS "C": aagttaagaggcaaga[70]) was used for all BKACE candidates and BPT, while either RBS "C" or an RBS with a higher predicted translation initiation rate (RBS "B": aacaaaatgaggaggtactgag[70]) was used upstream of MCR (see

Supplementary Data 2 for a list of plasmids used in this study, and Supplementary Data 3 for sequence maps).

### Growth media and growth characterization
Lysogeny broth (LB) media was used for routine culturing (1% NaCl, 0.5% yeast extract and 1% tryptone), supplemented with appropriate antibiotics for maintenance of plasmids. M9 minimal medium (Sigma-Aldrich, Cat. No. M6030) was used for all growth experiments, supplemented with trace elements (134 μM EDTA, 13 μM $FeCl_3$, 6.2 μM $ZnCl_2$, 0.76 μM $CuCl_2$, 0.42 μM $CoCl_2$, 1.62 μM $H_3BO_3$, 0.081 μM $MnCl_2$). Carbon sources were added as indicated in the main text, e.g. glycerol (20 mM), glucose (10 mM/20 mM), sodium L-lactate (20 mM), sodium formate. Lithium acetoacetate (Sigma-Aldrich, Cat. No. A8509) was used as source of acetoacetate. Sodium 1,3-$^{13}$C-acetoacetate was used for $^{13}$C-labeling (for synthesis, see above). Media containing acetoacetate were freshly prepared or stored at −20 °C to avoid variance in acetoacetate concentrations due to its long-term instability in aqueous solutions[118].

Minimal media containing formate (5 mM) were used to pre-culture strains, enabling CORE cycle-independent growth. Precultures were grown in 3 mL medium in 10 mL glass tubes under vigorous shaking at 37 °C. Antibiotics were added to precultures, when appropriate, but omitted for growth experiments. When acetoacetate was used as main carbon source in the final growth experiment, the preculture was performed in M9 + lithium acetoacetate (20 mM) + glycine (2 mM) + formate (5 mM) and streptomycin (where appropriate for maintenance of a plasmid). For all other experiments, the preculture was grown with glycerol as default carbon source, i.e., M9 + glycerol (20 mM) + glycine (2 mM) + formate (5 mM) and streptomycin (where appropriate). Streptomycin was omitted for untransformed or cured strains lacking a plasmid.

Cells from the preculture were pelleted and washed thrice in M9 medium without carbon source, prior to inoculation in 96-well plates (Nunclon Delta Surface, Thermo Scientific, Dreieich, Germany) at a starting $OD_{600}$ of 0.02. Wells were filled with 150 μL culture and covered with 50 μL mineral oil (Merck, Darmstadt, Germany) to avoid evaporation while allowing gas exchange. Aerobic growth was monitored in technical replicates (as indicated) at 37 °C in a BioTek Epoch 2 Microplate Spectrophotometer (BioTek, Bad Friedrichshall, Germany) by absorbance measurements (600 nm) of each well every ~10 min with intermittent orbital and linear shaking. Blank measurements were subtracted and $OD_{600}$ measurements were normalized to $OD_{600}$ values corresponding to a path length of 1 cm ("cuvette OD") by multiplying with a factor of 4.35, as established via a calibration curve for the instrument.

Maximal growth rates (and correspondingly, doubling times) were estimated for the mid-log phase using a custom MATLAB script.

### $^{13}$C isotopic labelling of proteinogenic amino acids
For isotope tracing, cells were cultured as triplicates (three separate cultures of the same clonal strain) in 14 mL glass culture tubes with 3 mL M9 medium containing the carbon sources. Cultures were inoculated to an $OD_{600}$ 0.01 and grown at 37 °C until an OD in late exponential phase, between 0.6–1.0. Then, a cell amount equivalent to 1 mL of culture with $OD_{600}$ 1 was pelleted (~$10^9$ cells), washed once with $ddH_2O$ and hydrolyzed in 1 mL hydrochloric acid (6 M) at 95 °C for 24 h. Subsequently, the acid was evaporated by heating at 95 °C under an air stream, and the hydrolyzed biomass was resuspended in 1 mL $ddH_2O$.

Relative quantification of isotopologues for amino acids of interest was performed using ultra-performance liquid chromatography high-resolution mass spectrometry (LC-MS). The chromatographic separation was performed on a Thermo Scientific Vanquish HPLC System using a ZicHILIC SeQuant column (150 × 2.1 mm, 3.5 μm particle size, 100 Å pore size) connected to a ZicHILIC guard column

(20 × 2.1 mm, 5 µm particle size) (Merck KGaA, Darmstadt, Germany) at a constant flow rate of 0.3 mL/min with mobile phase A being 0.1% formic acid in 99:1 water:acetonitrile (Honeywell, Morristown, New Jersey, USA) and phase B being 0.1% formic acid 99:1 acetonitrile:water (Honeywell, Morristown, New Jersey, USA) at 25 °C. The injection volume was 1 µL. The mobile phase profile consisted of the following steps and linear gradients: 0–8 min from 80 to 60% B; 8–10 min from 60 to 10% B; 10–12 min constant at 10% B; 12–12.1 min from 10 to 80% B; 12.1 to 15 min constant at 80% B. A Thermo Scientific ID-X Orbitrap mass spectrometer was used in positive mode with a high-temperature electrospray ionization (H-ESI) source and the following conditions: H-ESI spray voltage at 3500 V, sheath gas at 50 arbitrary units, auxiliary gas at 10 arbitrary units, sweep gas at 1 arbitrary units, ion transfer tube temperature at 325 °C, vaporizer temperature at 350 °C. Detection was performed in full scan mode using the Orbitrap mass analyzer at a mass resolution of 60 000 in the mass range 50 - 250 ($m/z$). Extracted ion chromatograms of the $[M+H]^+$ forms were integrated using the TraceFinder software (Thermo Scientific) applying a mass tolerance of 5 ppm. Amino acid standards (Merck KGaA, Darmstadt, Germany) were analyzed under the same conditions in order to determine expected retention times.

### Isolation of mutants capable of growing via the complete CORE cycle

Initial adaptive evolution was performed by manual serial passaging in 4 mL cultures of "acetoacetate medium" (M9 + 20 mM lithium acetoacetate+2 mM glycine+varying sodium formate concentrations). Initially, acetoacetate medium containing 200 µM sodium formate was inoculated with C1S-Aux+pCORE to an $OD_{600}$ of 0.01. Once growth had reached a stationary plateau above OD 0.5, the population was passaged at a dilution of 1:100 into a new culture of 4 mL, with a quarter of the previous formate concentration. This process was repeated twice, at which point the formate concentration in the medium had been decreased to 12.5 µM. From this passage, cells were transferred to three cultures lacking formate entirely. One of these cultures showed visible growth within 14 days and individual colonies (denoted Evo1, Evo1a, Evo1b, Evo1c in Supplementary Data 1) were isolated on solid "acetoacetate medium" (see above, with added 1.2% ultra-pure agarose). Growth of these clones was assayed in liquid "acetoacetate medium" lacking formate, identifying Evo1 as the best-performing isolate showing reproducible growth, thus used for further ALE.

### Long-term continuous culture evolution

Growth on acetoacetate as acetyl-CoA precursor of Evo1 bacteria (evolved from the C1S-Aux+pCORE strain by serial dilution) to close the CORE cycle was accelerated in a turbidostat conducted in a GM3 automated continuous culture device[48,58,59]. An Evo1 preculture (16 mL acetoacetate medium: M9 + 20 mM lithium acetoacetate+2 mM glycine, pH = 7.2) was injected in a GM3 growth chamber and grown at 37 °C. Every 10 min, the optical density of the culture was automatically measured and compared to a fixed threshold ($OD_{600}$ value of 0.4). When the measured OD exceeded the threshold, a 10% dilution pulse (1.68 mL) of fresh acetoacetate medium was injected by the fluidic system into the culture and the same volume of used culture was discarded. The dilutions ensured that the biomass in the vessel remained constant and that the bacteria grew at their maximal growth rate. Biofilm formation at the inner side of the growth chamber was counteracted by regular cycles of washing with 5 N NaOH and subsequent rinsing of the chamber with $H_2O$, thus enabling long term culturing periods. Once a week, a 1 mL sample of the culture was withdrawn and stored upon addition of DMSO (10% final) at −80 °C to constitute a stock of culture intermediates, from which evolved clones were isolated (e.g. Evo2/3/4).

### Whole genome re-sequencing

Genomic DNA was extracted using the NucleoSpin Microbial DNA purification kit (Macherey-Nagel, Germany), using cells harvested from an overnight culture in LB medium supplied with Streptomycin (to maintain the pCORE plasmid). Library preparation and short-read sequencing were performed by Novogene Inc. (Cambridge, UK). In brief, genomic DNA was randomly sheared into short fragments. The obtained fragments were end-repaired, A-tailed and further ligated with Illumina adapters. Fragments containing adapters were PCR-amplified, size-selected, and purified. Sequencing was performed on an Illumina NovaSeq X Plus platform to obtain 150 bp paired-end reads.

Mapping of reads and identification of mutations was performed using breseq v0.38.3[119] with two reference sequences: (1) the pCORE plasmid, and (2) a manually adjusted reference sequence based on the genome of *E.coli* MG1655 (GenBank accession NC_000913)[120]. The reference sequence used here contains several modifications compared to NC_000913, to reflect changes made during deletions and insertions in the genome (for a list of all sequence adjustments, see Supplementary Data 4).

### Plasmid curing

Plasmids were cured from isolated, evolved clones by propagation in non-selective minimal medium lacking antibiotics and supplemented with formate to enable loss of pCORE (M9 + 20 mM lithium acetoacetate + 2 mM glycine + 5 mM sodium formate). Clones which had lost the plasmid were identified by individually streaking on LB with and without Streptomycin. Plasmid loss was further verified by absence of a PCR product when amplifying with primers P01 and P02 (Supplementary Table 13).

### Lysate-based MCR activity measurements

Strains were grown to late exponential phase (OD 1.0) in 200 mL Terrific Broth (TB) medium with streptomycin, under vigorous shaking. Rich medium was chosen to enable comparison of the evolved strains with their non-evolved ancestor and plasmid-free C1S-Aux strain, both of which are incapable of fully CORE cycle-dependent growth. Cells from 200 mL culture were harvested by centrifugation, lysed in 50 mM HEPES pH 7.8, 150 mM KCl buffer via homogenization at 22,500 psi and cleared by ultracentrifugation at 100,000 g. MCR-activity was determined in a spectrophotometric assay as described above. Protein concentrations in clarified lysates were quantified using the Bradford assay (Quick Start Bradford 1x Dye reagent, BioRad). For higher sensitivity, absorbance was measured as 590/490 nm ratio[121].

### Flux balance analysis

To compare the CORE cycle variants to native plant photorespiration and previously proposed photorespiration bypasses, we performed stoichiometric modeling by applying flux balance analysis (FBA) with COBRApy (v0.20.0)[122], following a framework we used previously[33]. In brief, for each pathway, we calculated the consumption of ATP, NAD(P)H and reduced ferredoxins, as well as the required turns of the CBB cycle (including RuBisCO) to produce one unit of 3-phosphoglycerate. For this purpose, we constructed a simplified metabolic model consisting of the CBB cycle, specific reactions of each considered photorespiratory pathway, and generic/artificial cofactor regeneration and interconversion reactions (e.g., ADP + inorganic phosphate → ATP; NAD⁺ → NADH and NADH + NADP⁺ → NAD⁺ + NADPH; ATP + AMP → 2 ADP; oxidized ferredoxin → reduced ferredoxin). We note that transfer of electrons from NADH to NADPH, and vice-versa, is freely possible in this model without any ATP-investment via a generic transhydrogenase reaction (simulating photosynthetic production of NADPH). We assumed a RuBisCO carboxylation-to-oxygenation ratio of 3:1 (i.e. 25% of all RuBisCO reactions being oxygenations)[74,123,124]. To compare the yield of all pathways, we calculated their total required

ATP-equivalents to produce 3PG from $CO_2$, by using the conversions 1 NAD(P)H = 2.5 ATP[125,126] and 2 reduced ferredoxins = 1 NAD(P)H. The full code and the models can be found at the online repository Edmond (https://doi.org/10.17617/3.V8KJFV) within the "cost_comparison" directory. Model results are also provided in Supplementary Data 5.

FBA was also conducted to estimate the potential of the CORE cycle for synthetic autotrophy and $CO_2$-based bioproduction. We extended the CORE cycle with various formate assimilation pathways (see main text). We integrated the corresponding non-native reactions into the most updated *E. coli* genome-scale metabolic model (*i*ML1515)[127], with several curations and changes (see full code linked below). We used H2 as a generic electron/energy source (via hydrogenase: $H_2 + NAD^+ \rightarrow NADH + H^+$) and computed autotrophic yields of biomass and 12 bioproduction precursors[128] from $CO_2$. We used the CBB cycle (assuming RuBisCO carboxylation-to-oxygenation ratio of 3:1) as the benchmark to calculate the relative yields. The full script and models, including the individual changes, can be found at the online repository Edmond (https://doi.org/10.17617/3.V8KJFV) within the "bioproduction" directory.

## Reporting summary

Further information on research design is available in the Nature Portfolio Reporting Summary linked to this article.

## Data availability

Structural models for the BKACE15 apo-enzyme, enzyme with bound CoA and malonate, as well as enzyme with bound acetyl-CoA and acetoacetate were deposited to the Protein Data Bank in Europe under PDB accession 8RIO, 8RIP, and 9HNF, respectively. Raw data from next-generation sequencing has been deposited in the Sequence Read Archive (SRA) under accession PRJNA1124540. Source data are provided with this paper.

## Code availability

The code used for flux balance analysis has been deposited at Edmond [https://doi.org/10.17617/3.V8KJFV]. The code and reference sequences used to analyze sequencing data via breseq are also available from Edmond [https://doi.org/10.17617/3.NKWOPO].

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

## Acknowledgements

The authors are grateful to Beau Dronsella, Helena Schulz-Mirbach, Nico Claassens and Elad Noor for helpful discussions on the project and critical reading of the manuscript; to Peter Claus and Änne Michaelis for expert assistance with LC-MS measurements; to Hendrik Westedt for assistance with chemical synthesis; to Daniel Schindler and María Sánchez for helpful discussions on next-generation sequencing; to Frederik V. Schmidt and Christian Scholz for assistance with ICP-OES measurements. The authors acknowledge the assistance of the staff scientists of the beamlines P13 and P14 operated by EMBL Hamburg at the PETRA III storage ring (DESY, Hamburg, Germany). This work was supported by the Max Planck Society and the European Union H2020 Program (project GAIN4CROPS, grant agreement no. 862087). A.S. acknowledges funding from the International Max Planck Research School for Primary Metabolism and Plant Growth.

## Author contributions

A.B.-E. and A.S. conceived the study; A.B-E. and A.S. designed and analyzed $CO_2$-reduction cycles with contributions from S.B.; A.P., J-L.P. and V.P. designed and performed the initial in vitro screen of the BKACE collection; M.N. performed in vitro experiments on the reductive formamide pathway; D.G.M. designed and performed all other in vitro experiments, optimized the BKACE screening assay, synthesized and purified CoA-esters, MSA and 1,3-$^{13}C_2$-acetoacetate, purified enzymes and crystallized BKACE; D.G.M. and J.Z. collected and processed x-ray diffraction data, refined and analyzed crystal structures; A.S. designed and performed all in vivo experiments, designed ALE experiments, and analyzed genome sequencing data; M.B., V.D. and I.D. performed turbidostat evolutions; H.H., V.R. and H.P. contributed to strain engineering and in vivo experiments; A.S. and N.P. performed isotope tracing experiments and LC-MS analysis; H.H., E.S. and A.S. performed stoichiometric modeling; T.J.E., A.B-E. and M.H. supervised research and acquired funding; A.S., D.G.M. and T.J.E wrote the manuscript with contributions from all authors.

## Funding

## Competing interests

T.J.E. and A.B-E. are named inventors on the patent US10781456B2, describing photorespiration bypasses. The other authors declare no competing interests.
