## [Peer review file · Nature Communications]

Design and implementation of aerobic, ambient CO₂-reduction as an entry-point for enhanced carbon fixation

Corresponding Author: Professor Tobias Erb

Version 0:

Reviewer comments:

Reviewer #1

(Remarks to the Author)

Tobias Erb and colleagues describe a synthetic metabolic pathway (termed CORE cycle) for the direct reduction of CO₂ into formate and downstream molecules such as serine or methionine. The results are certainly noteworthy since this provides a new strategy to convert CO₂ under aerobic conditions and ambient CO₂ concentrations. It can be expected that the findings are highly significant to the field. Moreover and maybe more importantly, the study is a prime example about the enormous potential how to combine metabolic modules into a functional pathway that can be exploited for biotech purposes but may also be present (still undiscovered) in nature. The work fully supports the conclusions and claims. The authors' conclusions are based on multiple methods that have been performed very carefully. Especially the combination of theoretical pathway design, genetic engineering, smart ¹³C-labeling techniques, and enzyme evolution is quite unique and excellent. All methods and data are reported in considerable detail (in supplements).

(Remarks on code availability)

The mechanism for the BKACE reaction is plausible. Maybe, in future, the authors want to include ²H-labeling experiments. These could further provide some evidence for the reaction mechanisms, for example a possible back-transfer of H from the E-171 site to the products, but also in general for the whole pathway.

The manuscript is written very carefully that allows to follow the rationale of the authors. Very nice. In the text of the supplements, I have found a typo. Line 157 acetoacetate.

Reviewer #2

(Remarks to the Author)

The manuscript by Satanowski et al. describes the design and implementation of a synthetic pathway that reduces CO₂ to formate under aerobic conditions. The designed pathway uses a very unusual enzyme, BKACE, which is responsible for the transfer of acetyl residues of 3-ketocarboxylate to an acetyl residue of acetyl-CoA, resulting in the formation of acetoacetate, while CoA is attached to the corresponding truncated carboxylate (resulting in the formation of acyl-CoA). These enzymes are very slow catalysts that function, for example, in lysine fermentation or carnitine or catechol degradation. In the pathway designed by Satanowski et al., acetyl-CoA is carboxylated to malonyl-CoA and reduced to malonic semialdehyde, which reacts (in a BKACE reaction) with acetyl-CoA to form acetoacetate and formyl-CoA. Acetoacetate is then activated to acetoacetyl-CoA by CoA transfer from formyl-CoA or at the expense of ATP energy. In summary, the pathway catalyzes the aerobic ATP-dependent reduction of CO₂ to formate. When implemented in *E. coli*, this pathway is sufficiently active to meet the cellular demand for serine and C1-THF, demonstrating that the pathway can be functional in vivo, although it was responsible for only a small carbon flux in the engineered strain. The work includes biochemical characterization of heterologously produced proteins, structure determination, genomic modifications, ¹³C isotopic labelling analysis, laboratory evolution, and flux analysis. This is an excellent study and I have little to criticize.

In fact, the reduction of CO₂ to formate does not seem to be the most pressing problem of the envisaged formate bioeconomy. Nevertheless, the possibility of having more options for this critical step is important. It remains to be shown whether the fluxes through this pathway can be improved to allow autotrophic growth, and whether this growth will be truly efficient. Nevertheless, the manuscript is a very impressive example of a synthetic biology study using a wide range of state-of-the-art methods, and the amount of data in the manuscript is impressive. In fact, this is where I see the main problem of this manuscript: although it is very well written, it is so large that it is difficult to read due to the variety of aspects described. It

blurs the main point of the article. For example, I am not really sure whether it is necessary to discuss the variants of the design of the CO₂ reduction cycles other than the CORE cycle. Although they are interesting, they can be saved for future papers describing the implementation of these pathways. Here they distract from the main results of the manuscript. Also, I would suggest shortening the part on the genetic basis of evolved growth via the CORE cycle in the main text and moving it to the Supplementary Materials. In particular, the part on concatemers is interesting but does not add much to the main story.

l. 301: "overexpression": expression?

l. 425-427: "In terms of energetic efficiency, the CORE cycle requires 27 % less ATP and 16 % fewer reducing equivalents compared to native photorespiration and outperforms it by 25 % in terms of 3PG yield (Fig. S25, Table S8)": I am not sure I really understand how the CORE cycle can require less reducing equivalents compared to the natural photorespiration pathway, given that the CORE cycle involves the reduction of CO₂, whereas natural photorespiration results in the production of CO₂.

l. 537-543: I believe that the methods described is a slight modification of the Kroeger et al (2011) method (doi: 10.1016/j.ab.2010.11.046); please cite it.

l. 561, 564: please define the abbreviations first.

(Remarks on code availability)

Reviewer #3

(Remarks to the Author)

The manuscript introduces the CORE cycle, a synthetic metabolic pathway that converts CO₂ to formate. The CORE cycle employs a reduction-first strategy, converting CO₂ initially into a reduced C1-compound, which is subsequently condensed into multi-carbon compounds. A critical challenge in this field is establishing an oxygen-tolerant reduction-first CO₂ fixation. The authors claim to have designed several pathways energized by ATP hydrolysis, splitting the task of CO₂ reduction into multiple enzymatic steps with favorable thermodynamic driving forces. After designing ten different hypothetical pathways, the authors focused on the CORE cycle and solved the crystal structures of BKACE15, which mediates a new-to-nature reaction essential in the CORE cycle. They also demonstrated that the CORE cycle supports *E. coli* growth by supplementing C1 metabolism and serine biosynthesis from CO₂. Overall, this study is well-designed, well-performed, and represents a significant advancement in the field of CO₂ reduction. However, there are several concerns which need to be addressed.

1. In the Introduction section, studies on NADH-dependent formate dehydrogenase should be revisited and included. Recent studies on FDH have shown successful conversion of CO₂ to formate using synthetic electron mediators such as ethyl viologen under aerobic conditions with high CO₂ concentrations.
2. Regarding Table S11, the second structure (PDB 8RIP) was solved in the P212121 space group, distinct from the other two structures, although their cell parameters are quite similar. Please explain how the space group was determined to ensure the correctness of the structure solutions.
3. For PDB 8RIQ, the Rmerge value is uncharacteristically high considering the high-resolution data (~2 Å). This could be due to an incorrect space group or the inclusion of problematic diffraction data.
4. It appears that malonate is not the substrate of BKACE15; rather, malonate semialdehyde is the correct substrate. Why was malonate used? The additional oxygen in malonate could potentially cause artifacts in substrate binding, as suggested by the poor fit of malonate in Fig. S13A's Fo-Fc map.
5. Concerning PDB 8RIQ, it is unclear why both acetoacetate and acetyl-CoA were used simultaneously, especially since acetoacetate is a product while acetyl-CoA is a substrate.
6. In Fig. S13, it is unclear how the metal density was modeled with Zn²⁺ ions. Metal ion identities are difficult to distinguish based solely on electron densities. An anomalous difference map of Zn²⁺ should be shown to clarify its identity.
7. Based on structural observations that each active site has only one entry channel for both substrates, the authors hypothesized that malonate semialdehyde must enter the active site before acetyl-CoA can bind. However, it could also be possible that acetyl-CoA enters the active site first. This mechanism could be experimentally verified through kinetic studies, which the authors are well-positioned to conduct.
8. The manuscript states that the product, acetoacetate, can inhibit BKACE15 activity, hypothesizing it acts as a competitive inhibitor with MSA and acetyl-CoA. Enzyme products can also bind to allosteric sites, potentially affecting enzyme activity. To clearly elucidate the inhibition mechanism of acetoacetate on BKACE15, kinetic studies should be conducted to measure changes in K_m and V_{max} values before and after acetoacetate treatment.
9. The finding that BKACE15 forms a homotetramer with four active sites is intriguing. Is there any evidence of cooperativity among these active sites?
10. Figure S23 presents homology models of AtoA. It would be interesting to use AlphaFold to predict the structures of mutants.
11. The K_m values of BKACE15 seem quite high, in the sub-millimolar range. Could the authors estimate the cellular concentrations of acetyl-CoA and MSA when the CORE cycle is implemented in *E. coli*?
12. The motivation behind the structural studies on BKACE15 was to understand its catalytic behavior. It is assumed that the authors aimed to elucidate the structures of BKACE15 in various states to achieve this goal. However, there is limited analysis of the structure-function relationship in their work. Comparing the active sites of BKACE15 with those of homologous structures could provide valuable insights into its catalytic activity from a structural perspective. Additionally, a potential reaction mechanism might be suggested based on the coordination of substrates or products.
13. Compared to extensive studies, the discussion section lacks sufficient insights derived from this research. Additional discussions are needed to further improve the CORE cycle and to explore what can be learned to realize other hypothetical

pathways (Figs. S1-S10).

(Remarks on code availability)

Version 1:

Reviewer comments:

Reviewer #2

(Remarks to the Author)

I am satisfied with the revision and will be happy to see the manuscript published.

(Remarks on code availability)

Reviewer #3

(Remarks to the Author)

The manuscript introduces the CORE cycle, a synthetic metabolic pathway that converts CO₂ to formate, representing a significant advancement in the field of CO₂ reduction. This reviewer suggested several straightforward experiments to strengthen the manuscript, particularly focusing on the structural features of BKACE15. Despite the general importance of the manuscript, a mechanistic understanding of BKACE15 is crucial for its impact. Hence, the authors focused their detailed studies on determining crystal structures.

The authors generally acknowledged the points raised to enhance the manuscript. However, instead of conducting additional experiments—such as generating an anomalous difference map and performing kinetic studies to measure K_m and V_{max} values for the inhibition mechanism—which are feasible and would strengthen the manuscript, the authors chose to explain their reasoning and tone down their original claims. Additionally, the authors' explanations for the space group issues and high R_{merge} value are not sufficiently convincing. This is a critical concern as it pertains to the manuscript's credibility. A more thorough analysis is required to support their conclusions.

(Remarks on code availability)

Version 2:

Reviewer comments:

Reviewer #3

(Remarks to the Author)

The authors have made sufficient revisions to the manuscript, and all of my concerns have been addressed. I recommend that the manuscript be published.

(Remarks on code availability)

Yes, the code provides clear instructions for installing and running the application.

REVIEWER COMMENTS

Reviewer #1 (Remarks to the Author):

Tobias Erb and colleagues describe a synthetic metabolic pathway (termed CORE cycle) for the direct reduction of CO₂ into formate and downstream molecules such as serine or methionine. The results are certainly noteworthy since this provides a new strategy to convert CO₂ under aerobic conditions and ambient CO₂ concentrations. It can be expected that the findings are highly significant to the field. Moreover and maybe more importantly, the study is a prime example about the enormous potential how to combine metabolic modules into a functional pathway that can be exploited for biotech purposes but may also be present (still undiscovered) in nature. The work fully supports the conclusions and claims. The authors' conclusions are based on multiple methods that have been performed very carefully. Especially the combination of theoretical pathway design, genetic engineering, smart ¹³C-labeling techniques, and enzyme evolution is quite unique and excellent. All methods and data are reported in considerable detail (in supplements).

We thank the reviewer for the positive assessment of our work.

Reviewer #1 (Remarks on code availability):

The mechanism for the BKACE reaction is plausible. Maybe, in future, the authors want to include ²H-labeling experiments. These could further provide some evidence for the reaction mechanisms, for example a possible back-transfer of H from the E-171 site to the products, but also in general for the whole pathway.

We thank the reviewer for this advice. We indeed plan to conduct ²H-labeling experiments in the future to further corroborate the proposed mechanism for the interesting carbon-carbon rearrangement of BKACE, and trace back the hydrogen transfer of the enzyme during the reaction of ²H-malonate semialdehyde with acetyl-CoA. We hope that for this study, which centers on the design and realization of the CORE cycle, our conclusions on the mechanism are sufficient.

The manuscript is written very carefully that allows to follow the rationale of the authors. Very nice. In the text of the supplements, I have found a typo. Line 157 acetoaceate.

Thank you! We corrected the typo.

Reviewer #2 (Remarks to the Author):

The manuscript by Satanowski et al. describes the design and implementation of a synthetic pathway that reduces CO₂ to formate under aerobic conditions. The designed pathway uses a very unusual enzyme, BKACE, which is responsible for the transfer of acetyl residues of 3-ketocarboxylate to an acetyl residue of acetyl-CoA, resulting in the formation of acetoacetate, while CoA is attached to the corresponding truncated carboxylate (resulting in the formation of acyl-CoA). These enzymes are very slow catalysts that function, for example, in lysine fermentation or carnitine or catechol degradation. In the pathway designed by Satanowski et al, acetyl-CoA is carboxylated to malonyl-CoA and reduced to malonic semialdehyde, which reacts (in a BKACE reaction) with acetyl-CoA to form acetoacetate and formyl-CoA. Acetoacetate is then activated to acetoacetyl-CoA by CoA transfer from formyl-CoA or at the expense of ATP energy. In summary, the pathway catalyzes the aerobic ATP-dependent reduction of CO₂ to formate. When implemented in *E. coli*, this pathway is sufficiently active to meet the cellular demand for serine and C1-THF, demonstrating that the pathway can be functional in vivo, although it was responsible for only a small carbon flux in the engineered strain. The work includes biochemical characterization of heterologously produced proteins, structure determination, genomic modifications, ¹³C isotopic labelling analysis, laboratory evolution, and flux analysis. This is an excellent study and I have little to criticize.

We thank the reviewer for the very positive evaluation and concise summary of our work. Indeed, the carbon flux demand in our selection system was rather low when providing our proof of principle. However, we are confident that the CORE cycle will be able to satisfy higher cellular demands in future work – especially when extrapolating from known catalytic efficiencies of the individual reactions used to construct the cycle.

In fact, the reduction of CO₂ to formate does not seem to be the most pressing problem of the envisaged formate bioeconomy. Nevertheless, the possibility of having more options for this critical step is important. It remains to be shown whether the fluxes through this pathway can be improved to allow autotrophic growth, and whether this growth will be truly efficient. Nevertheless, the manuscript is a very impressive example of a synthetic biology study using a wide range of state-of-the-art methods, and the amount of data in the manuscript is impressive. In fact, this is where I see the main problem of this manuscript: although it is very well written, it is so large that it is difficult to read due to the variety of aspects described. It blurs the main point of the article. For example, I am not really sure whether it is necessary to discuss the variants of the design of the CO₂ reduction cycles other than the CORE cycle. Although they are interesting, they can be saved for future papers describing the implementation of these pathways. Here they distract from the main results of the manuscript. Also, I would suggest shortening the part on the genetic basis of evolved growth via the CORE cycle in the main text and moving it to the Supplementary Materials. In particular, the part on concatemers is interesting but does not add much to the main story.

We agree with the reviewer that the concept of a “formate bioeconomy” can take advantage of non-biological routes to produce formate from CO₂ using renewable energy, e.g. electrochemical approaches (see e.g. Satanowski & Bar-Even *EMBO Rep* 2020). With the CORE cycle, we specifically wanted to contrast these physical approaches with a biological solution that is suitable for organisms (e.g., photosynthetic (micro)organisms & plants), that cannot easily be supplied with exogenous formate and/or have access to a surplus of energy & reducing power (e.g., light, H₂). Indeed, the future will tell, whether and how much the CORE cycle can contribute to autotrophic growth – as the successful implementation of artificial CO₂-fixation pathways is (still) a challenge.

We also appreciate the reviewer’s advice on increasing readability and conciseness of our manuscript. We carefully considered the suggestion to omit the discussion on the other pathway designs. However, we feel that it would be important to provide the reader with a more general view on the potential design space (e.g., as in Bar-Even et al. *PNAS* 2010; Schwander et al. *Science* 2016), which allows to highlight commonalities and differences between the different designs. Also providing the information, which of the designs could not be realized yet in this study might be valuable to the synthetic biology community and eventually stimulate enzyme engineers to develop the missing enzyme activities.

I. 301: “overexpression”: expression?

Agreed! We changed the wording.

I. 425-427: “In terms of energetic efficiency, the CORE cycle requires 27 % less ATP and 16 % fewer reducing equivalents compared to native photorespiration and outperforms it by 25 % in terms of 3PG yield (Fig. S25, Table S8)”: I am not sure I really understand how the CORE cycle can require less reducing equivalents compared to the natural photorespiration pathway, given that the CORE cycle involves the reduction of CO₂, whereas natural photorespiration results in the production of CO₂.

We thank the reviewer for this question, which prompted us to explain our result better in the Supplement. Indeed, the CORE cycle (as part of a CO₂-assimilating photorespiratory bypass) would seem to require more energy and reducing power compared to “native photorespiration”. Intuition would suggest that the CORE cycle requires more resources, as it must invest energy and reducing equivalents to assimilate additional carbon, while “native photorespiration” simply converts 2-phosphoglycolate to glycerate involving CO₂ release.

However, one needs to take into account that native photorespiration requires a **second** molecule of 2-phosphoglycolate (as a “sacrificial” source of methylene-THF via glycine decarboxylation; see Fig. S25), which in turn must be (re)generated via three turns of the entire CBB cycle, including one Rubisco oxygenation reaction (in Table S8, we show the calculated numbers of CBB cycle turnovers for each of the photorespiration pathways discussed), which adds additional energy and reducing equivalent cost to native photorespiration.

In contrast, such additional turns of the CBB cycle are not required for the CORE cycle, because the pathway assimilates CO₂. We used flux-balance analysis to stoichiometrically compare the different solutions and accurately balance all inputs and outputs for the production of one molecule of 3-phosphoglycerate from three molecules of CO₂. Thus, our calculation compares not only the photorespiratory routes alone, but also takes into account the required resource investments for flux through the CBB cycle. Please note that this calculation is standard in the field and has been used in earlier calculations on native and synthetic photorespiration (e.g., Trudeau et al. *PNAS* 2019; Scheffen et al. *Nature Catalysis* 2021).

We hope these additional notes clarify the reason why the CORE cycle outperforms native photorespiration in terms of energetic efficiency. We have highlighted these points in a footnote in Table S8, but would like to refrain from discussing these points in the main text.

I. 537-543: I believe that the methods described is a slight modification of the Kroeger et al (2011) method (doi: 10.1016/j.ab.2010.11.046); please cite it.

We thank the reviewer for pointing this out and added the missing citation.

I. 561, 564: please define the abbreviations first.

We have now added the full enzyme names to this section of the methods, explaining the abbreviations.

Reviewer #3 (Remarks to the Author):

The manuscript introduces the CORE cycle, a synthetic metabolic pathway that converts CO₂ to formate. The CORE cycle employs a reduction-first strategy, converting CO₂ initially into a reduced C1-compound, which is subsequently condensed into multi-carbon compounds. A critical challenge in this field is establishing an oxygen-tolerant reduction-first CO₂ fixation. The authors claim to have designed several pathways energized by ATP hydrolysis, splitting the task of CO₂ reduction into multiple enzymatic steps with favorable thermodynamic driving forces. After designing ten different hypothetical pathways, the authors focused on the CORE cycle and solved the crystal structures of BKACE15, which mediates a new-to-nature reaction essential in the CORE cycle. They also demonstrated that the CORE cycle supports *E. coli* growth by supplementing C1 metabolism and serine biosynthesis from CO₂. Overall, this study is well-designed, well-performed, and represents a significant advancement in the field of CO₂ reduction. However, there are several concerns which need to be addressed.

Thank you for the encouraging feedback. We hope to have addressed all open points in our revision!

1. In the Introduction section, studies on NADH-dependent formate dehydrogenase should be revisited and included. Recent studies on FDH have shown successful conversion of CO₂ to formate using synthetic electron mediators such as ethyl viologen under aerobic conditions with high CO₂ concentrations.

We thank the reviewer for bringing up this point. As the reviewer suggested, we actually refer to the possibility of using formate dehydrogenases (FDH) for CO₂ reduction and cite one study in the Introduction that we found most relevant in this context (lines 69-73, reference 21). This study confirmed the requirement for elevated CO₂ concentrations and strongly altered cellular redox homeostasis when using FDH. As the reviewer noted, the use of elevated CO₂ concentrations is true for most, perhaps all, *in vitro* studies investigating CO₂ reduction by FDHs (which are often achieved by adding high bicarbonate concentrations, e.g. 100 mM). Moreover, while artificial, low-redox electron donors such as (methyl/ethyl) viologen can be used in cell-free, *in vitro* studies, they are inherently incompatible for use within aerobic, living cells. For these reasons, we defined specific design goals for our synthetic pathways, i.e., conversions of CO₂ to formate that are (i) feasible at ambient/atmospheric CO₂ concentrations; (ii) feasible inside living cells, including regeneration of the utilized electron donors; and (iii) oxygen-tolerant.

Taking into account that reviewer #2 suggested to further shorten our manuscript, we would prefer to keep the current focus of our introduction on CO₂ reduction systems that are applicable *in vivo* and at ambient CO₂-concentrations and not further delve into details of different FDH systems.

2. Regarding Table S11, the second structure (PDB 8RIP) was solved in the P212121 space group, distinct from the other two structures, although their cell parameters are quite similar. Please explain how the space group was determined to ensure the correctness of the structure solutions.

The space groups were determined by *XDS* as well as *Pointless*, and additionally confirmed by *Xtriage* with no suspicious values or properties. Although twinning is potentially possible in both types of space groups, the L-test did not indicate twinning for any of the datasets. We are convinced that the respective space groups are correct. The similarity in unit cell dimensions appears to be by chance.

3. For PDB 8RIQ, the Rmerge value is uncharacteristically high considering the high-resolution data (~2 Å). This could be due to an incorrect space group or the inclusion of problematic diffraction data.

This dataset has a very high multiplicity of over 26 because it is a combined dataset of two crystals. This is the reason for a slightly higher R_{merge} value. You may notice, however, that the R_{pim} value (redundancy-independent) is in a typical range.

4. It appears that malonate is not the substrate of BKACE15; rather, malonate semialdehyde is the correct substrate. Why was malonate used? The additional oxygen in malonate could potentially cause artifacts in substrate binding, as suggested by the poor fit of malonate in Fig. S13A's Fo-Fc map.

Malonate semialdehyde (MSA) is unfortunately not very stable and starts to polymerize quickly in aqueous solutions, which makes co-crystallization practically impossible. We tried several different crystallization conditions and soaked crystals with different mixtures of CoA, acetyl-CoA, MSA, acetoacetate, as well as potential reaction intermediate analogs. In this case, we found a condition with 200 mM malonate, where we could obtain convincing density for CoA after soaking. But we were not able to replace the malonate in the active site with any other tested substrate/product, likely because of the high concentration of malonate in the particular condition.

We believe that malonate serves as a non-reactive substrate analog and would argue that the fit into the simulated annealing omit map at a sigma of over 2 is actually quite good. The electron density of the zinc ion and the carboxyl oxygens of malonate overshadow the density of the central carbon atom to some extent, but we can observe a clear connection of the density between the two carboxyl groups.

5. Concerning PDB 8RIQ, it is unclear why both acetoacetate and acetyl-CoA were used simultaneously, especially since acetoacetate is a product while acetyl-CoA is a substrate.

We assumed that acetoacetate would serve as a non-reactive substrate, if acetyl-CoA would be used as a co-substrate. Note that acetoacetate can serve as substrate in a productive "back-reaction", e.g. if formyl-CoA was used as co-substrate (forming malonate semialdehyde and acetyl-CoA). However, formyl-CoA is unstable in aqueous solutions, difficult to synthesize and purify (half-life of few minutes). We therefore tested acetoacetate and acetyl-CoA as a combination of substrates in order to obtain good electron densities that are unambiguous and not blurred by reactions taking place within the crystals.

6. In Fig. S13, it is unclear how the metal density was modeled with Zn²⁺ ions. Metal ion identities are difficult to distinguish based solely on electron densities. An anomalous difference map of Zn²⁺ should be shown to clarify its identity.

In addition to previous literature describing Zn²⁺ as the bound ion in BKACE (PMID: 21632536 and PMID: 24240508), we found that BKACE candidates produced in *E. coli* and purified in absence of Zn²⁺ showed activity only upon supplementation of Zn²⁺ to the enzyme assays. Addition of Mg²⁺ and Mn²⁺ resulted in no such activities. For this reason, we always included ZnCl₂ in our protein purification buffers, as well as in all activity assay buffers. Since none of the crystallization conditions contained other added divalent metal ions, we concluded that the metal ion in the coordination center must be Zn²⁺.

Additionally, in our structures, we observed a tetrahedral coordination sphere of Zn²⁺ formed by two histidines, aspartate and acetoacetate/malonate (see Figure S13), which is unique to Zn-dependent enzymes to the best of our knowledge (Dudev & Lim, *J Phys Chem B* 2001). In absence of substrate, an interaction with water is also conceivable (e.g. in Fig. S13A where CoA is bound without an acyl-moiety). In contrast, Mg²⁺ forms an octahedral coordination sphere and needs 6 ligands to neutralize its positive charge.

In response to the reviewer's comment, we now added more details to the Methods section that explain, why we chose to model Zn²⁺ instead of another metal ion (see sections 'Spectrophotometric measurements' and 'Crystallization and structure determination of BKACE15').

7. Based on structural observations that each active site has only one entry channel for both substrates, the authors hypothesized that malonate semialdehyde must enter the active site before acetyl-CoA can bind. However, it could also be possible that acetyl-CoA enters the active site first. This mechanism could be experimentally verified through kinetic studies, which the authors are well-positioned to conduct.

We thank the reviewer for this critical point. Our structure with bound acetyl-CoA shows that this compound completely fills out the space of the active site entry channel. Therefore, a mechanism, where MSA binds first and acetyl-CoA second, appears much more likely than an alternative binding sequence. To clarify this in the manuscript, we have now updated Figure 2F by showing a surface representation of acetyl-CoA in magenta that fills the space of the entry channel completely. We have also added the following sentence in the main text:

“Each active site has only one entry channel for both substrates that appears fully occupied upon acetyl-CoA binding (Fig. 2f).“

8. The manuscript states that the product, acetoacetate, can inhibit BKACE15 activity, hypothesizing it acts as a competitive inhibitor with MSA and acetyl-CoA. Enzyme products can also bind to allosteric sites, potentially affecting enzyme activity. To clearly elucidate the inhibition mechanism of acetoacetate on BKACE15, kinetic studies should be conducted to measure changes in K_m and V_{max} values before and after acetoacetate treatment.

We thank the reviewer for pointing this out. Given that acetoacetate can bind the same active site and could even serve as a substrate for the reverse reaction, a competitive inhibition seemed more likely to us. However, we agree with the reviewer that the observed inhibition may additionally stem from allosteric effects. Since our study focusses on the *in vivo* implementation of the pathway, the exact mode of inhibition seems less crucial than the fact that inhibition is observed in general. We have therefore now omitted the word “competitive” from the manuscript and reduced our claim to state that acetoacetate is found to be an inhibitor of BKACE15 in general. We have adapted the text as follows:

Line 180: *“However, we also observed that the enzyme was inhibited by its product acetoacetate with a half-maximal inhibitory concentration (IC50) of 1.45 ± 0.16 mM (Fig. 2d).“*

Line 194: *“This hypothesis is further corroborated by the observation that acetoacetate is an inhibitor for the reaction with MSA and acetyl-CoA.“*

9. The finding that BKACE15 forms a homotetramer with four active sites is intriguing. Is there any evidence of cooperativity among these active sites?

Following this suggestion, we have revisited our structural data but could not detect any evidence of cooperativity. An overlay of the four subunits shows virtually identical structures. However, we note that all crystallization experiments were conducted under substrate-saturating conditions while cooperative effects usually appear under sub-saturating conditions. Therefore, we cannot rule out cooperativity completely. The Michaelis-Menten kinetics that we performed also did not show a sigmoidal curve that is typical for enzymes exhibiting cooperativity.

10. Figure S23 presents homology models of AtoA. It would be interesting to use AlphaFold to predict the structures of mutants.

We thank the reviewer for this suggestion. Indeed, we considered using AlphaFold, especially given its higher accuracy compared to other tools. However, we aimed not only to predict the structure of the mutant variants but also to “see” how substrates are bound within the active site. The capabilities of AlphaFold models to reliably predict substrate binding are still limited, therefore we decided for SWISS-MODEL, which offers the possibility to select a specific structure template. We chose a template with a covalently bound reaction intermediate. We believe this approach produces a more accurate prediction of the active site than AlphaFold. We have now added a more detailed description of the AtoA modelling to the Methods section and refer to it in the caption of Figure S23.

11. The K_m values of BKACE15 seem quite high, in the sub-millimolar range. Could the authors

estimate the cellular concentrations of acetyl-CoA and MSA when the CORE cycle is implemented in *E. coli*?

According to literature, CoA esters are typically found in the high micromolar to low millimolar range in microbial cells. For instance, acetyl-CoA is typically found at concentrations around 0.6 mM (Bennett et al. *Nat Chem Biol* 2009; own experiences), which is in line with the kinetic parameters of BKACE15 (Km for acetyl-CoA ~0.1 mM). In our experience, *E. coli* tolerates free aldehydes in the high micromolar to low millimolar range, which is still compatible with the Km of BKACE15 for MSA (~0.8 mM). As a comparison, the Km of the enzyme 3-hydroxypropionate dehydrogenase is ~0.1 mM for MSA, which is a bit lower, but still within an order of magnitude of the value for BKACE15. Quantifying absolute concentrations of free aldehydes in cell extracts / cells is quite complicated due to their inherent reactivity with amines and thiol groups.

12. The motivation behind the structural studies on BKACE15 was to understand its catalytic behavior. It is assumed that the authors aimed to elucidate the structures of BKACE15 in various states to achieve this goal. However, there is limited analysis of the structure-function relationship in their work. Comparing the active sites of BKACE15 with those of homologous structures could provide valuable insights into its catalytic activity from a structural perspective. Additionally, a potential reaction mechanism might be suggested based on the coordination of substrates or products.

Thank you for this comment. In Supplementary Text 2 (noted in line 199 in the main text), we describe a structure-function comparison of the active site of BKACE15 with other BKACE homologues, and also discuss the reaction mechanism. We now expanded this section by an additional paragraph about previously proposed subgroups of the Pfam family to which BKACE15 belongs. Based on these comparisons and our structures, we propose an alternative reaction mechanism that is shown in Figure S15 and elaborated in Supplementary Text 2 (which is referred to in line 199 in the main text).

13. Compared to extensive studies, the discussion section lacks sufficient insights derived from this research. Additional discussions are needed to further improve the CORE cycle and to explore what can be learned to realize other hypothetical pathways (Figs. S1-S10).

We thank the reviewer for the suggestion to expand the discussion section. However, given the other reviewer's advice to rather shorten the article, we would prefer to keep the discussion concise and not add too many speculations about future improvements or challenges in the further implementation of the CORE cycle. We (and others) recently published reviews that summarize general lessons for the *in vivo*-implementation of synthetic pathways (e.g., Schulz-Mirbach et al. *Metab Eng* 2024), which we cite throughout this manuscript and that might be better suited to inform the reader.

REVIEWER COMMENTS

Reviewer #2 (Remarks to the Author):

I am satisfied with the revision and will be happy to see the manuscript published.

We thank the reviewer for the positive assessment of our work.

Reviewer #3 (Remarks to the Author):

The manuscript introduces the CORE cycle, a synthetic metabolic pathway that converts CO₂ to formate, representing a significant advancement in the field of CO₂ reduction. This reviewer suggested several straightforward experiments to strengthen the manuscript, particularly focusing on the structural features of BKACE15. Despite the general importance of the manuscript, a mechanistic understanding of BKACE15 is crucial for its impact. Hence, the authors focused their detailed studies on determining crystal structures.

The authors generally acknowledged the points raised to enhance the manuscript. However, instead of conducting additional experiments—such as generating an anomalous difference map and performing kinetic studies to measure K_m and V_{max} values for the inhibition mechanism—which are feasible and would strengthen the manuscript, the authors chose to explain their reasoning and tone down their original claims.

We thank the reviewer for their valuable feedback.

Regarding the suggestion to generate an anomalous difference map to determine the bound metal ion, we note that we have already provided very strong experimental evidence that Zinc is the active site metal. We showed that the enzyme purified in absence of metals is functionally inactive and becomes only active when adding Zinc to the buffer. Additionally, there is consensus in the literature that these enzymes use Zinc as active site metal (e.g., doi: 10.1074/jbc.M111.253260).

However, to strengthen the argument that the enzyme is Zinc-dependent, we provide additional data in our second revision. For time reasons, we decided not for additional crystallization trials and X-ray diffraction measurements. Instead, we conducted an ICP-OES (Inductively Coupled Plasma Optical Emission Spectroscopy) analysis to determine the metal content of BKACE15, definitely confirming Zinc (Zn) as the bound metal in a 1:1 ratio (Zinc:active site). We added these new results as Figure S28 and refer to this result in line 191 of the main text:

“As previously reported for the BKACE from Candidatus Cloacamonas acidaminovorans, a close homolog of BKACE15, each active site coordinates a single Zn²⁺ ion (Bellinzoni et al., 2011). We confirmed a 1:1 zinc binding ratio in BKACE15 using Inductively Coupled Plasma Optical Emission Spectroscopy (ICP-OES, Fig. S28).”

In respect to the inhibitory mechanism, we agree with the reviewer's comment that further kinetic studies would offer additional insights. However, we believe that such information is rather peripheral to the content of the paper. We agree that the apparent inhibition by acetoacetate is an important observation, but think that the *precise mode of inhibition* is not central to the findings of our study, nor crucial for future applications of BKACE15 (as we show in our manuscript).

Nevertheless, we followed the reviewer's suggestion and conducted additional LC-MS assays. These assays, involving varying concentrations of MSA and acetoacetate, allowed us to further explore the inhibition mechanism. The results of these experiments are now presented in Figure S27, with a corresponding update in line 182 of the manuscript:

“... we also observed that the enzyme was inhibited by its product acetoacetate with a half-maximal inhibitory concentration (IC_{50}) of 1.45 ± 0.16 mM (Fig. 2d, Fig. S27).”

Please note that performing detailed kinetic studies of this enzyme reaction is highly challenging due to several technical limitations: Firstly, no continuous spectrophotometric or high-throughput assay is available for this system. Moreover, both the substrate MSA (high chemical reactivity) and the product formyl-CoA (rapid hydrolysis) are exceedingly unstable in solution, imposing severe constraints on experimental design. The additional assays we performed were not only laborious, time- and resource-intensive, but also subject to variability due to the inherent instability of these compounds.

In our added LC-MS assays, we measured combinations of four MSA concentrations with four acetoacetate concentrations, respectively. The data clearly rules out uncompetitive inhibition, and suggests that competitive and/or noncompetitive mechanisms of inhibition are at work (Fig. S27). The ability of acetoacetate to bind to the active site simultaneously with acetyl-CoA (as indicated by our crystal structures and described in the main text) strongly supports a competitive mechanism of inhibition. However, additional noncompetitive inhibition – even though highly unlikely – at an allosteric site cannot be completely excluded based on our data. A definitive discrimination between these two mechanisms would require measurements at saturating MSA concentrations (and in excess to acetoacetate), which are impractical given our previous observation that MSA (a highly reactive aldehyde) leads to decreased BKACE15 activity at concentrations above 5 mM (as described in the main text).

In summary, the inherent challenges associated with kinetic studies of this reversible, multi-substrate reaction, coupled with the instability of key reactants and products, limit the feasibility of a more comprehensive mechanistic study of BKACE15 inhibition by acetoacetate.

We hope the reviewer appreciates the additional efforts we undertook during the revision and the complexity of the system under investigation.

Additionally, the authors' explanations for the space group issues and high R_{merge} value are not sufficiently convincing. This is a critical concern as it pertains to the manuscript's credibility. A more thorough analysis is required to support their conclusions.

As suggested by the reviewer, we re-analyzed the structural data for BKACE15 with bound acetyl-CoA and acetoacetate. Specifically, we reprocessed the data using only one of the two measured X-ray diffraction datasets, thereby reducing the multiplicity of the data. As a result, the novel dataset has a lower R_{merge} value of 0.1114 (0.841), compared to the previous value of 0.148 (1.442). The resolution of the new dataset remains almost identical to the previous one: 27.69 - 2.1005 (2.16 - 2.1005) Å compared to the earlier value of 27.69 - 2.05 (2.16 - 2.05) Å. We deposited the corresponding structure in the PDB (9HNF) and updated table S11 in the Supplementary Materials accordingly.

Regarding the space group assignments, we used multiple (standard) programs like XDS, Pointless, Aimless, and Xtriage to ensure their validity. Additionally, the L-test confirmed that twinning was not present in any of the datasets. While both space groups theoretically permit twinning, no evidence of twinning was observed. To ensure the robustness of our conclusions, we also attempted to force each dataset into the respective other space group, but this approach failed to produce useful results and did not allow to solve the phasing problem. This strongly supports the correctness of the assigned space groups.

The similarity in unit cell dimensions for the datasets solved in different space groups appears to be coincidental. The unit cells and space groups are neither equivalent nor is twinning indicated.

We have taken these additional steps to rigorously verify our findings and are convinced that the assigned space groups are correct. We trust that this thorough re-analysis and the additional steps taken to validate the structural data provide sufficient reassurance of the accuracy and reliability of the reported structures.

REVIEWERS' COMMENTS

Reviewer #3 (Remarks to the Author):

The authors have made sufficient revisions to the manuscript, and all of my concerns have been addressed. I recommend that the manuscript be published.

We thank the reviewer for their time and effort in reviewing our manuscript, and for positively recognizing our revisions.

Reviewer #3 (Remarks on code availability):

Yes, the code provides clear instructions for installing and running the application.